# Brief communication: Glacier thickness reconstruction on Mt. Kilimanjaro

Catrin Stadelmann[1], Johannes Jakob Fürst[1], Thomas Mölg[1] and Matthias Braun[1]

[1]Institute of Geography, Friedrich-Alexander University Erlangen-Nürnberg, Erlangen, 91058, Germany

*Correspondence to*: Catrin Stadelmann (catrin.stadelmann@fau.de)

**Abstract.**

Glaciers on Kilimanjaro are unique indicators for climatic change in the tropical mid-troposphere of Africa, but their disappearance seems imminent. A key unknown is their present ice thickness. Here, we present thickness maps for the Northern Icefield (NIF) and Kersten Glacier (KG) with mean values of 26.6 m and 9.3 m respectively in 2011. In absence of direct
measurements on KG, multi-temporal satellite information was exploited to infer past thickness values in areas that have become ice-free and therefore allow glacier-specific calibration. In these areas, KG is unrealistically thick in the existing consensus estimate of global glacier ice thickness.

## 1 Introduction

Tropical glaciers at high elevations are unique climate indicators for the tropical mid-troposphere (e.g. Kaser 2001, Mölg et
al. 2009). As one of few tropical locations with still existing glaciers, Mt. Kilimanjaro, a stratovolcano with an elevation of 5895 m.a.s.l., is located in East Africa close to the Tanzania-Kenya border (3°04' S / 37°21' E) (Fig. 1, overview). In addition to the very high elevation, the free-standing nature of the mountain causes the glacier on top of the summit to be directly exposed to tropospheric flows at higher altitudes, minimizing the forcing of local climate on the glacier and creating a unique opportunity to study the mid-troposphere climate.

The modern glacier recession on Kilimanjaro has been well documented and mapping approaches have shown that from an estimated ice extent of 11.4 km² in 1912, only 1.76 km² remained in 2011, constituting a severe 85% reduction in glacier area (Cullen et al. 2013). While glaciological research on Kilimanjaro has focused on mapping glacier area and retreat (e.g. Cullen et al. 2013), as well as quantifying the mass and energy balance (Mölg et al. 2003, 2008, 2009), the research on the ice thickness of different glaciers on Kilimanjaro has been comparably sparse (Bohleber et al. 2017). However, in light of severe glacier
recession, an assessment of current glacier thickness is important to better determine future recession. A recent effort was made to reconstruct the distributed ice thickness for all glaciers outside Antarctica and Greenland using a consensus of up to 5 models (Farinotti et al. 2019). This estimate generated ice thicknesses for Northern Icefield (NIF) and Kersten Glacier (KG) using ensembles of 2 and 3 models, respectively. The consensus estimate produces a similar mean ice thickness of 21.5 m for NIF, which is in fair agreement with the observations by Bohleber et al. (2017), considering that the consensus was not

informed by local thickness observations(Farinotti et al. 2019). Moreover, the recently observed separation of KG into two
fragments (e.g. Landsat 5 scene 2011-08-22; Image courtesy of the U.S. Geological Survey) is not in agreement with the high
thickness values found in the consensus estimate.

Here, we present the first well constrained thickness maps for KG and NIF using a mass conserving reconstruction approach
introduced in Fürst et al. (2017) that readily assimilated thickness measurements (Section 3.3). In two different experiments
we test the influence of varying input of ice thickness observations for the glacier state of 2000, where we rely on surface mass
balance (SMB) data from a physically-based  model developed by Mölg et al. (2008, 2009; Section 3.1) and digital elevation
data with global coverage (Shuttle Radar Topography Mission; SRTM; USGS), pursuing a new calibration strategy that uses
multi-temporal satellite information on geometric changes in absence of observational ice thickness data on KG. These
resulting thickness estimates are then compared to the consensus estimate (Farinotti et al. 2019). In a third experiment, we
combine the very high resolution digital elevation model KILISoSDEM (0.5 m ground resolution; Sirguey et al. 2014) with
the calibration strategy from the previous 2000 experiments to produce a best estimate for the 2011 glacier state.

## 2 Data

To apply the distributed surface mass balance (SMB) model (Sect. 3.1; Mölg et al., 2008, 2009) and the thickness
reconstruction (Sect. 3.3), the following input data was used: climate data measured by the automatic weather station (AWS)
located on KG (Section 3 c in Mölg et al. 2009), digital elevation information from the SRTM digital elevation model (DEM)
from 2000 and the KILISoSDEM from 2012 (Sirguey et al. 2014), the RGI6.0 glacier outlines from 21 February 2000 (RGI
Consortium 2017), as well as digitized outlines based on a Landsat 5 image from 22 August 2011. The change in surface height
between 2000 and 2011 was found by differencing a merge of two TanDEM-X radar images from 2011 (28 January 2011, 4
April 2011; Suppl. Fig. 2; for details on the processing refer to Braun et al., 2019) and the SRTM DEM.

The central plateau area of NIF drains westward into two glaciers, Drygalski Glacier in the south and Credner Glacier in the
North. In anticipation of a future separation of NIF, we redefine Credner Glacier (CG) to comprise the northern part of NIF
(Fig. 1). Ice thickness measurements on Kilimanjaro are limited to NIF, where three ice cores were drilled to bedrock in 2000,
with lengths from 49.0 m (C1) to  50.9 m (C2) and 50.8 m (C3) (Thompson et al. 2002; Fig. 1 for borehole locations). In
addition, ground penetrating radar (GPR) profiles from September 2015 (Fig. 1) were collected by Bohleber et al. (2017).
Using a kriging interpolation and the KILISoSDEM, the authors estimated the mean thickness to be between $21.2 \pm 1$ m and
$27 \pm 2$ m. For the anticipated reconstruction in 2000 and 2011, the GPR thickness measurements for NIF are adjusted by
linearly extrapolating the above-mentioned elevation change information to the elapsed time between the DEM date and
acquisition date of the thickness measurements.

## 3 Methods

### 60 3.1 Mass balance modelling

The mean annual climatic surface mass balance fields were generated using version 2.4 of the distributed, physically-based mass balance (MB) model by Mölg et al. (2008, 2009), driven by meteorological input from the aforementioned AWS (Suppl. Fig. 1). The full MB model has already been calibrated and validated for KG. For the application on NIF, surface meltwater is not expected to run off but rather refreeze over the very flat plateau areas (Mölg & Hardy, 2004). To properly reproduce these

conditions on NIF, we revised the model code so that refreezing of meltwater can occur on a bare ice surface with a slope angle below 5 degrees (not captured before).With these changes, the model is capable of reproducing the surface height changes observed by a Sonic Ranger mounted to the AWS.

### 3.2 Margin thickness generation

For KG, no in-situ thickness measurements are available. Therefore, multi-temporal DEM and glacier outline information is

used to infer past ice thickness. First, glacier retreat is delineated from outline information in 2000 and 2011 (Fig. 1 hatched area). In the currently ice-free area, contemporaneous elevation changes (2000-2011) then give information on past ice thickness. Positive values, which indicate a height gain in the TanDEM-X layer, were removed as a height gain outside the 2011 glacier extent implies an increase in glacier thickness from 2000 to 2011, which is unlikely. In total we removed 92 of 602 grid cells with a mean height gain of 0.19 m/a for NIF and 14 of 254 grid cells with a mean height gain of 0.25 m/a for

KG.

### 3.3 Ice thickness reconstruction

Here, we apply an inverse method to infer the glacier ice thickness by assimilating surface observations and ground-truth thickness measurements. A detailed description of this two-step data assimilation, of which we only used the first model step as surface velocities were not available, can be found in Fürst et al. (2017). The reconstruction approach is based on the

principle of mass conservation and computes a glacier-wide flux field from the difference between the surface mass balance (Section 3.1) and contemporaneous elevation changes. The flux solution is converted into thickness values using the Shallow Ice Approximation (SIA; Hutter, 1983). This conversion involves the ice-viscosity parameter $B$, which is a-priori unknown. This parameter stems from assuming a Glen-type flow law, linking deviatoric stresses to strain rate components $\dot{\varepsilon}_{ij}$ via the effective viscosity $\eta = 0.5\ B^{-1/n}\ \dot{\varepsilon}^{(1-n)/n}$. Here, $\dot{\varepsilon}$ is the second invariant of the strain rate tensor (for further information on the

equation see Pattyn 2003) and $n=3$. After the flux solution is obtained, $B$ is calibrated at locations where thickness measurements are available. This point-information is then expanded to the entire glacier domain using a Natural Neighbor Sibsonian Interpolation, resulting in a spatially variable field. Before interpolating the $B$ values from each measurement

location to the entire glacier basin, an average value is prescribed along the glacier outline to avoid spurious extrapolation effects.

To account for different availability of thickness measurements, Fürst et al. (2017) conducted experiments withholding 1% - 99% of the available point measurements on several test geometries on Svalbard. Aggregate errors typically exceed 10-20% of the mean glacier ice thickness when most measurements are withheld but error values quickly reduce as measurements become available. Between the two end-member experiments (1% and 99%), volumes of the test geometries differ by at most 10%. Considering input uncertainties from the DEM and the SMB fields, sensitivity tests revealed that ice-volume differences

remain below 5% (only shown for the ice-cap geometry). For more details on associated uncertainties and input sensitivities, we refer the interested reader to Fürst et al. (2017).

For the reconstruction in 2000, a nominal mesh resolution of 25 m was chosen. With the higher DEM quality in 2011, the resolution was iteratively increased from 25, via 10 and 5, to 2 m. The processing was conducted separately for NIF and KG. To smooth the surface slope during reconstruction, we use a coupling length parameter (introduced in Fürst et al. 2017), which

is defined as a multiple of the local ice thickness. In this way, flux streamlines become less erratic and their alignment increases. For KG, the parameter is set to 1, a typical value for valley glaciers (Kamb & Echelmeyer, 1986). For NIF, it had to be reduced so that the steep elevation increase at the vertical ice cliffs is depicted in the thickness field. A compromise value of 0.3 was chosen to still guarantee sufficient smoothing of the flux streamlines.

### 3.4 Experimental Setup

The general strategy is to reconstruct a thickness field for KG and NIF by combining SMB, elevation changes and glacier geometry with in-situ measurements of ice thickness for two points in time, namely 2000 (Experiment 1 and 2) and 2011 (Experiment 3). In Experiment 1, we reconstructed the glacier state for 2000 with the generated margin thickness data (Section 3.2) for both NIF and KG. As KG is rather small, we expect a homogeneous ice viscosity. In Experiment 2, we therefore decided to simply average the point information on ice viscosity and use a constant viscosity value over the entire glacier basin.

In this way, lateral thickness values are no longer reproduced but spurious spatial viscosity variations stemming from the generic margin data are suppressed. For NIF in Experiment 2, we chose to use the thickness measurements from Bohleber et al. (2017) as input, to check how observational data influences the glacier-wide ice thickness in comparison to only using margin thickness information. In Experiment 3, the aim is to benefit from the 2011 KILISoSDEM showing very high resolution. For NIF, the reconstruction can still be calibrated by GPR measurements from Bohleber et al. (2017) acquired in

central areas. For KG, the retreat information falls outside the ice-covered domain in 2011. Therefore, we use the mean viscosity information as inferred for the reconstruction in year 2000 (Experiment 2). The KILISoSDEM is further exploited to investigate the resolution influence. Table 1 summarizes the three different experimental setups (Tab. 1).

## 4 Results

Results show generally larger ice thickness for NIF than for KG in all three experiments. For Experiment 1 (Fig. 2 A), KG
shows thickest ice of up to 15 m at the flat plateau parts of KG. For the central areas on the mountain flank thickness values
show a mean of 6.2 m and locally reach up to 7.5 m, with patches of thinner ice towards the glacier margins. NIF is up to 40
m thick in its center, decreasing towards the glacier margins and towards CG and has a mean ice thickness of 13.7 m. At the
borehole locations C1, C2 and C3 ice thicknesses of Experiment 1 are 19.9 m, 23.9 m and 36.6 m thinner, respectively (see
Suppl. Table 1).

Results from Experiment 2 show a similar thickness pattern on KG (Fig. 2B). For NIF, the magnitude differs significantly.
Now one large part of NIF's flat area and two smaller parts of CG exceed 40 m. Moreover, the ice thickness in the steeper
western areas of NIF and CG has increased by a factor of 2. The mean ice thickness also increases to 23.4 m. At the borehole
locations C1, C2 and C3 ice thicknesses of Experiment 2 are 4.4 m, 8.3 m and 26.1 m thinner, respectively (see Suppl. Table
1). Concerning the GPR surveys from Bohleber et al. (2017), the thickness map of NIF (Fig. 2D) largely reproduces these
measurements. Turning to the consensus estimate map (Farinotti et al., 2019), larger discrepancies prevail (Fig. 2E), especially
towards the eastern part. KG shows a similar thickness, but the ice body on the mountain flank becomes thicker in the central
parts. As before, the thickest ice patch remains on the plateau. The mean ice thickness with 6.9 m is very similar to
Experiment 1. In Experiment 3 (Fig. 1) we now move forward in time to 2011. Here, KG is split into two parts and shows an
ice thickness of up to 10 m at the flat top part and most of the slope being between 5 and 7.5 m thick. KG's mean ice thickness
is 9.3 m. NIF's thickness distribution is similar to Experiment 2, with the thickest areas of over 40 m at its flat part on the
plateau. For NIF, the decrease in thickness is less noticeable than the lateral retreat and decrease of glacier area. The mean ice
thickness of NIF in Experiment 3 is 26.6 m. At the three ice-core locations, the thickness mismatch remains comparable to
Experiment 2 (see Suppl. Table 1), with a mean relative absolute difference of 26%. This value is rather large and exceeds
inferred error estimates for the majority of glacier on Svalbard (Fürst et al., 2017). Here, we want to use it as a rough guideline
for the overall uncertainty of the 2011 reconstruction.

## 5 Discussion

We will first discuss the reconstructions for the year 2000 (Fig. 2): Generally, our experiments produce results with a higher
difference in thickness magnitude between KG and NIF as compared to the consensus thickness map (Fig. 2C; results from
Farinotti et al. 2019). For KG, no ice thickness measurements are available, and it is uncertain to what extent the generated
thicknesses along the glacier margin (Section 3.2) are useful to inform the reconstruction. We find that the margin values result
in a spatially varying viscosity field, which is transmitted into the ice thickness field (Experiment 1; Fig. 2 A). As no strong
viscosity variations are expected for the small KG, a second run was conducted with constant viscosity (Experiment 2; Fig. 2
B). Results indicate a thick central flow unit, as one might expect for a steep glacier, as well as a smoother ice thickness
distribution, with higher thickness in the center of the glacier and thinning towards the margins. In the absence of ground truth

data, it is unclear, which thickness field is more plausible. However, as the thickness of most glaciers on Earth is unsurveyed, the use of margin thickness information, generated from outline differences enabled a local glacier-specific viscosity tuning which might be preferential to an empirical temperature relation (Huss and Farinotti, 2012). The consensus map shows a similar thickness pattern as Experiment 2. The most notable difference is found for the thickness magnitude of KG. For the consensus estimates, thickness values exceed 35 m both for the flat top part and the central steep slope part. The consensus

mean thickness of 27.1 m, is more than twice as large as in our reconstruction. Since there are no actual thickness observations for KG, it is not certain that the ice was only up to 15 m thick in the year 2000. However, KG split in two parts by 2011. The separation line follows a contour line just below the plateau. Mean elevation changes between 2000 and 2011 of -0.64 m/yr suggest that not more than 7 m of ice was present in 2000. With 35 m ice in this area, the consensus estimate seems too large. NIF's peculiar geometry poses a challenge and it is difficult to reconstruct the ice thickness distribution using generic thickness

observations around the margin (Experiment 1, Fig. 2A). The ice is much too thin in the interior (Fig. 2 A), which underestimates the ice core lengths from Thompson et al. 2002 by 48, 52 and 71% for the core locations C1, C2 and C3 respectively. When the interior GPR measurements are used as model input (Experiment 2; Fig. 2 B), differences decrease to 10, 17 and 53%. Large mismatch values, especially for borehole C3, might as well be explained by the very flat plateau. Therefore, ice motion is expected to be rather slow. Stagnant and flat areas are challenging for a reconstruction based on ice-

flow and Fürst et al. (2017) show that uncertainties in the reconstructed thickness values significantly increase towards the ice divide of an ice cap. They further show that measurements along divides and ridge areas are most valuable to constrain the reconstruction approach used here. Although GPR measurements are available on the NIF plateau, we expect that uncertainties increase quickly away from these measurements. This can partly explain the mismatch with C3. Turning to the consensus estimate the mismatch is significantly larger with relative underestimations of 34%, 38% and 72% for boreholes C1, C2 and

C3. The complex topography posed a similar challenge for the models participating in the consensus, especially because no thickness observations were considered for Mt. Kilimanjaro. This is also reflected in the similar mean ice thicknesses, which are 27.1 m for the consensus estimate (Farinotti et al. 2019) and 23.4 m for Experiment 2. The results for KG and NIF point towards glacier margin data as thickness information being more useful in dynamically more active areas, such as KG, instead of more stagnant and flat areas, such as the flat plateau area of NIF.

Experiment 3 repeats the reconstruction for the year 2011 at a very high resolution. The general distribution of ice thickness is barely affected by the increase in resolution. This stability under resolution increase is assuring and illustrates the effects of inherent smoothing via the coupling length parameter that scales with the thickness. For NIF however, resolution is key, and the cliff geometries are much better imprinted in the final thickness map. Further experiments with 10 and 5 m model resolution (not shown) showed barely any difference in thickness distribution, verifying this effect. The mean ice thickness for KG and

NIF have increased in comparison to Experiment 2 from 6.9 m to 9.3 m and from 23.4 m to 26.6 m, respectively. As observed elevation changes do not support an increase, remaining explanations comprise model resolution, outline differences and DEM quality. Resolution can be excluded based on results from a 25 m reconstruction in 2011 (not shown). Concerning the 2011 outlines, some internal ice-free areas (on both NIF and KG), present in the RGI, could not be confirmed from the coarse

Landsat imagery, resulting in thicker ice. The four holes in the RGI outlines for KG and NIF either stay ice-free in the 2011
outline, connect to the lateral ice-free area or are located very close to the 2011 margin. Therefore, the reconstruction accounts
for these indirectly. Bohleber et al. (2017) show more even smaller ice-free spot in the western part of NIF, which could not
be confirmed from the coarse Landsat imagery thus explaining thicker ice there. The quality difference between SRTM and
KILISoSDEM is certainly also a contributing factor explaining part of the larger thickness values.

Finally, we want to briefly discuss the reconstruction approach, used here, with respect to other strategies for inferring
distributed thickness information. The Ice Thickness Models Intercomparison eXperiments (IMTIX; Farinotti et al. 2017)
concluded that as long as no thickness measurements are available, no single strategy generally outperforms the others. In this
case, an ensemble result from multiple models is preferable. Yet here, observations are either available or are inferred from
multi-temporal satellite imagery. Measurement availability was used in the global consensus estimate to infer performance
scores for the participating models (Farinotti et al., 2019) and the approach by Fürst et al. (2017) was attributed the highest
value. Yet, with regard to applications on individual geometries as for Kilimanjaro, comparable results, as presented here,
might well be attainable with various approaches. Regarding input requirements, approaches based on the perfect plasticity
assumption are least exigent, only requiring information on the ice geometry (e.g. Frey et al., 2014).

## 6 Conclusion & Outlook

This study has a multi-disciplinary character as we apply modelling approaches for glacier surface mass balance, infer remotely
sensed elevation changes and utilize available information in a data assimilation. The aim of the assimilation is to accurately
determine the thickness and distribution of ice for NIF and KG on Mt. Kilimanjaro.

In the reconstructions for 2000 we assessed the utility of this margin thickness information in constraining glacier thickness
by comparing our reconstructions to the recent global consensus estimate (Farinotti et al. 2019). For Kersten Glacier, we report
significantly smaller thickness values as compared to the consensus estimate, which is shown to be inconsistent with the
observed glacier separation between 2000 and 2011. Our reconstructions for 2011 show mean ice thicknesses of 9.3 m for KG
and 26.6 m for NIF. A comparison of modelled thickness to the ice core lengths (Thompson et al. 2002) results in a mean
relative absolute error of 26%. For NIF, our reconstruction (Experiment 2) and the consensus estimate both show a very similar
mean ice thickness, which is surprising as the consensus estimate was not informed by any thickness measurements.

As ice thickness observations were not available for KG, the reconstruction approach was calibrated with past thickness values
inferred from multi-temporal satellite information in areas that became ice-free in the last decade. In absence of ground truth
data on KG, it remains unclear if the retreat information is best used spatially distributed or as a bulk average. Yet we can state
that increased quality and resolution of more recent DEMs are key for capturing sharp transition zones (e.g. cliffs). The lateral
glacier retreat information seems less utile as central ice thickness is strongly underestimated. Reasons for this worse
performance might be the complex topography and the dynamic inactivity of NIF. We therefore speculate that thickness
information from retreat is most useful in areas that have been dynamically more active in the past. But we also have to

acknowledge that the unique glacier settings on Mt. Kilimanjaro are certainly not ideal for this first assessment of utilizing glacier retreat information to allow a glacier-specific calibration of thickness reconstruction approaches and should be tested elsewhere to verify our results.

To conclude, glacier retreat is palpable all around the planet and it will continue in the future (e.g., Hock et al., 2019). As time progresses, the suggested strategy to infer past ice thickness values from multi-temporal satellite information will produce an increasing wealth of calibration data. Moreover, the approach is readily transferable and provides a means for glacier-specific calibration of reconstruction approaches on regional or even global scales.

**Acknowledgement**

CS received primary funding from the project BR2105 /14-1 within the DFG Priority Program "Regional Sea Level Change & Society". JJF was funded by the German Research Foundation (DFG) under grant number FU1032/1-1. Results presented in this publication are based on numerical simulations conducted at the high-performance computing center of the Regionales Rechenzentrum Erlangen (RRZE) of the University of Erlangen-Nürnberg. We would also like to thank Nicolas Cullen and Pascal Sirguey (both at University of Otago, NZ) for constructive discussions and for providing the KILISoSDEM.

**Author contribution**

CS led writing of the manuscript, in which she received support from all authors. The research aims and setup were developed in regular discussion with JJF, TM and MB.

**Competing interests**

The authors declare no competing interests.

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

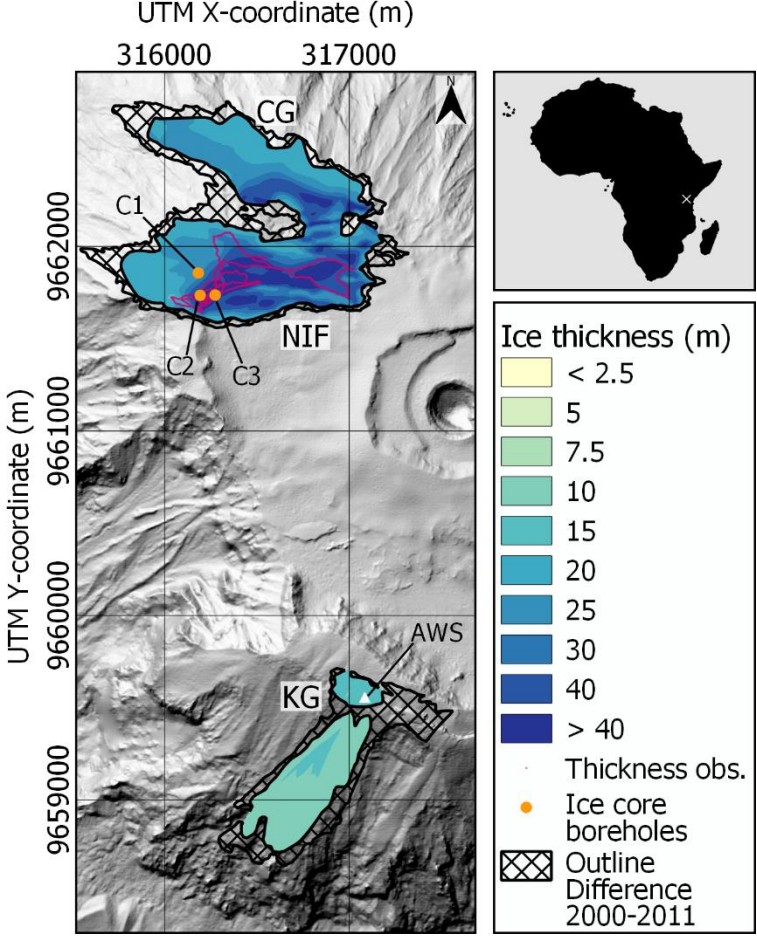

**Figure 1: Reconstructed ice thickness (m) for Northern Icefield (NIF) and Kersten Glacier (KG) for the year 2011, based on thickness observations (NIF) and mean viscosity (KG) with a model resolution of 2 m (Experiment 3). The magenta path on NIF represent the GPR ice thickness measurements by Bohleber et al. (2017). Orange dots (C1-C3) indicate the drill locations of the ice cores from Thompson et al. (2002). The AWS location from Mölg et al. (2009) is marked by the white triangle. Background: KiliSoSDEM hillshade. The overview map depicts the location of Mt. Kilimanjaro (white cross) on the African continent.**

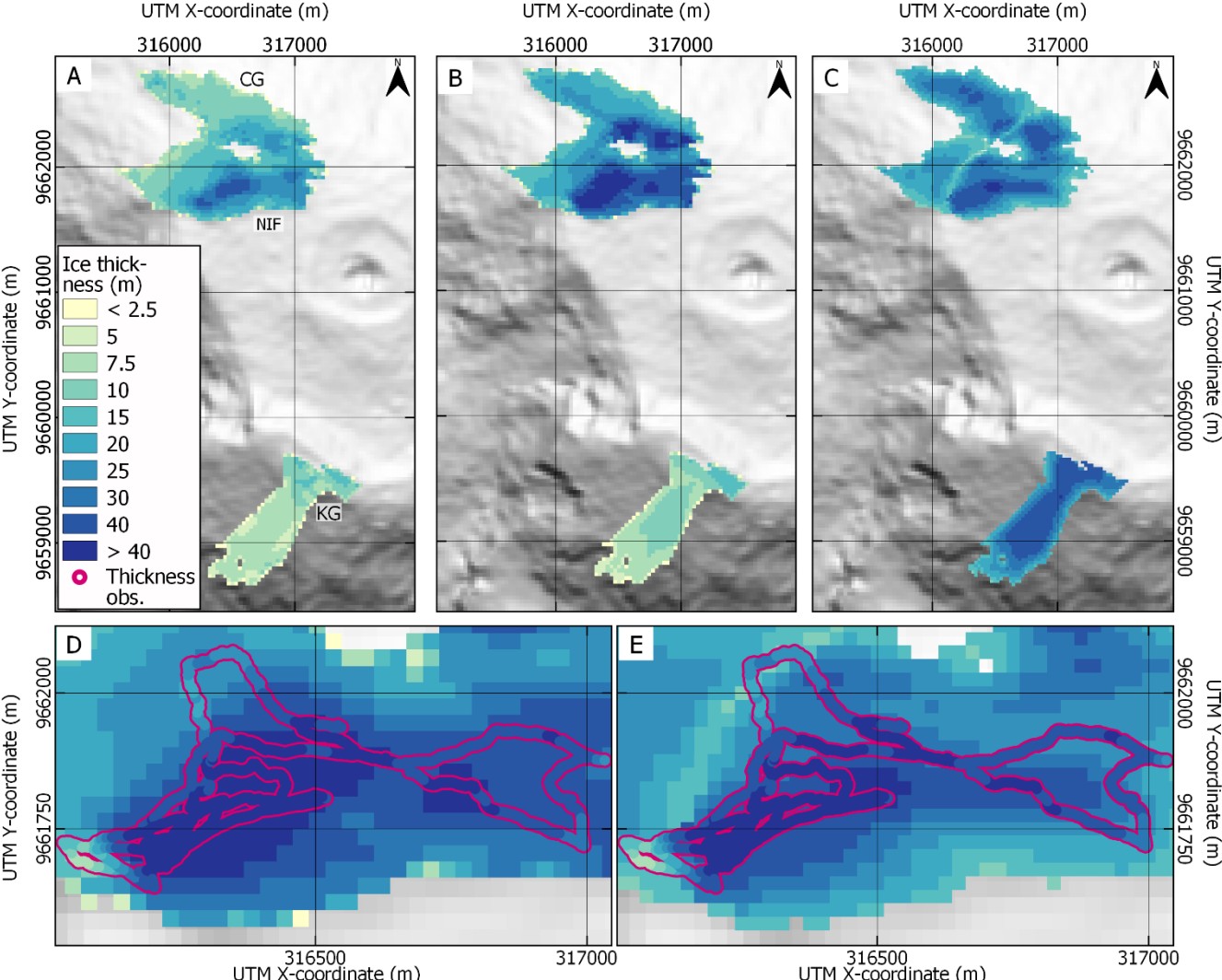

Figure 2: Reconstructed ice thickness (m) for Northern Icefield (NIF) and Kersten Glacier (KG) for the year 2000. Panel (A) shows results for Experiment 1 making exclusive use of past thickness information in nowadays ice-free areas. Panel (B) presents results from Experiment 2, which uses a bulk viscosity inferred from the retreat information on KG, whereas for NIF, the reconstruction is only calibrated by in-situ GPR measurements (see Table 1).As comparison panel (C) depicts the composite ice thickness from Farinotti et al 2019. Panel (D) shows a closeup of NIF, overlying the thickness map of Experiment 2 with the GPR thickness measurements (magenta contour, showing measured values in the same colourbar) by Bohleber et al. (2017). Panel (E): same as Panel (D) but showing the consensus thickness map (Farinotti et al., 2019; cf. Panel C). Background: SRTM DEM hillshade.

**Table 1: Experimental setup of the ice thickness reconstruction (Section 3.4).**

|  | Representing the glacier state for the year | Thickness input KG | Thickness Input NIF | Surface Elevation Information (year of acquisition) | Glacier Outlines (acquisition date) | Mean ice thickness KG (m) | Mean ice thickness NIF (m) |
|---|---|---|---|---|---|---|---|
| **Experiment 1** | 2000 | Generated margin thicknesses | Generated margin thicknesses | SRTM DEM (2000) | Randolph Glacier Inventory 6.0 (2000/02/21) | 6.2 | 13.7 |
| **Experiment 2** | 2000 | Mean viscosity | Observations from Bohleber et al. (2017) | SRTM DEM (2000) | Randolph Glacier Inventory 6.0 (2000/02/21) | 6.9 | 23.4 |
| **Experiment 3** | 2011 | Mean viscosity | Observations from Bohleber et al. (2017) | KILISoSDEM (2012) | Digitized from Landsat 5 Image (2011/08/22) | 9.3 | 26.6 |