# Peer review of "Brief communication: Glacier thickness reconstruction on Mt. Kilimanjaro"

_The Cryosphere, 2019_

## Referee Comment (RC1) · Anonymous Referee #1 · 24 Feb 2020

**A) Summary**

The manuscript presents estimates for the ice thickness distribution for the glaciers on Mount Kilimanjaro. The estimates refer to the years 2000 and 2011, and are based on a combination of in-situ observations, past ice thickness reconstructions derived for areas that are now ice free and a numerical, ice-flux based approach. The paper seems to have two main points. For one, the available global-scale ice thickness estimates seem to have overestimated the ice thickness for one of the two investigated glaciers. For another, the idea of using a combination of past and present digital elevation models (DEMs) to derive ice thickness observations that can be passed to ice-flux estimation approaches is suggested to hold promise for future applications. The paper has a general good quality, and the findings are certainly worth conveying to the larger

audience. Slight improvements seem necessary in the way that individual details are presented. The discussion section could benefit from a somewhat more substantial revision.

B) Major comments

1) In various instances, I had difficulty in following the details of the presentation. Often, I believe that it comes down to slightly revising the wording. In other occasions, I felt that some methodological information was missing. I understand that a "brief communication" has not the same amount of space available as regular papers, but still I think that some improvement can be done. I hope that the line-by-line comments can be useful to address the various cases.

2) Somehow, I was left in doubt on how the available Ground Penetrating Radar (GPR) measurement enter the game. They are briefly mentioned in the Data section (L. 48), do not show up in the Methods, and re-appear in the Discussion (L. 145). In particular, clarification is required for what the mentioned "assimilation" (L. 33 and 145) actually entails. As the manuscript stands now, no information is provided, and that should be rectified.

3) I had some reasonably hard time in following the discussion. I found it particularly hard to keep track of the many comparisons done for the two glaciers, targeting at the three Experiments performed in the work itself, the two (or three?) models used in the consensus estimate, and the two available sources of in-situ observations (boreholes and GPR data). To me, it would seem natural to show a figure depicting the various model results along the available GPR transects. Since both surface DEM and thickness are available for any of the various results, all information required to generate such a plot seems available. Most likely, this would help the readers to better grasp the main outcome of the discussion which, as far as I understand, rather focuses on the performance of the consensus estimate than on the results of the manuscript itself?

4) An important point of discussion that seems to have been missed is that ice thick-
ness estimation approaches as used in this study require the investigated glaciers to have some ice flux. Otherwise, the main idea behind the approaches somewhat breaks down. This point is skimmed in the Conclusions & Outlook section (L. 183) but would probably deserve some space in the Discussion section as well. May it help to explain some of the discrepancies noted between model results and observations?

5) The last few sentences of the Conclusions & Outlook (L. 190-198) seem the paper's strongest and most valuable point. Shouldn't these implications be highlighted in the abstract as well?

C) Minor comments

1) There are several undefined acronyms, including, amongst other, SRTM at L. 35, MB at L. 61, TDX at L. 66.

2) I could not follow the logics exposed at L. 61-63. According to the sentence, the surface mass balance model applied in the study was "slightly altered" because (sic) "it was never tested for Kersten Glacier before". I imagine that the model was actually tested by the authors before altering it, and that the matter is only one of wording?

3) At L. 69-72 the authors state that they removed all positive elevation differences from the analysis because such positive changes are "unlikely" to happen. The issue is that this removal apparently affects some 15% of the area of the Northern Icefield, which calls for some more detail. For example: What is the spatial distribution of these removed cells? Is it completely scattered, suggesting random noise, or is it clustered, indicating that the signal might be real after all? What is the confidence in the individual DEMs? Etc.

4) I was not able to follow L. 90-94. A "coupling length parameter" is introduced without further explanation (I assume the definition is found in Fuerst et al. 2017, which is ok) and, as far as I understand the wording, is first said to control how the surface DEM is "imprinted in the thickness field" (I'm not entirely sure what this means) and later said

to control the "smoothness" on not further specified "flux streamlines". I don't want to exclude that the wording makes perfect sense to a reader familiar with the details of Fuerst et al (2017) but I think that some additional words of explanation will help the majority of the readership.

D) Line-by-line comments

A (rather long, I apologise) set of line-by-line comments is found in the annotated document, attached to this review. The comments provided above are contained therein as well.

Please also note the supplement to this comment:
https://www.the-cryosphere-discuss.net/tc-2019-310/tc-2019-310-RC1-supplement.pdf

---

## Referee Comment (RC2) · Anonymous Referee #2 · 24 Feb 2020

**1. General comments**

In this study, the authors estimate the ice thickness of Northern Icefield and Kersten Glacier on Mt. Kilimanjaro in 2000 and 2011 using the ice thickness approach by Fürst et al (2017). Three different "experiments" are conducted to estimate the ice thickness, which either improves or are within the estimates from previous studies. The study makes good use of the few available observations, and the method and results are generally sound and interesting. I know this is a brief communication, but there are some key pieces of information that are missing within the data and model descriptions which I think are necessary to understand the manuscript, and the results would benefit from a discussion of uncertainties. In addition, the conclusion needs to be rewritten, as it does not seem to fit with the rest of the paper. In general, the manuscript would

also benefit from an increase in specificity and clarity, as I often had trouble following the text. I hope the technical comments are usefull for improving this.

2. Specific comments

L40: Location of the AWS is not on Figure 1. Also, for how long a period has the AWS measured and what components are measures (and with what uncertainties)?

L48-49: This is not quite correct. The thickness estimate was created from the GPR by doing kringing interpolation, and it is an estimate from the whole area, not just the flat central part. Later in the manuscript (L 169), you seem to use the estimate that Bohleber created from the DEM, so I would mention that result here too. E.g. "In addition, ground penetrating radar (GPR) profiles from September 2015 (Fig. 1) were created by Bohleber et al. (2017). Using a kringing interpolation and the KILISoSDEM, the authors estimated the mean thickness to be between $21.2 \pm 1$ m and $27 \pm 2$ m."

L55-56: The Bohleber estimate is from 2015 and the consensus estimate is from 2000, so that should also contribute to the differences.

L39-56: For most of the observations you do not provide uncertainty estimates

L61: You have a point measurement in one location. How to you get distributed mass balance maps from one point on one glacier? And do you use the mean SMB from 2005-2013 for the 2000 and 2011 estimation? If you do, you should mention this as a possible source of uncertainty (and if you don't, how do you find the 2000 SMB?).

L61-65: Again, what do you use as forcing for NIF if the AWS is on KG? And how do you use the sonic ranger to test refreezing on NIF if it is mounted on KG? In addition, you should mention the sonic ranger in the data section and not only in the methods.

L87: What method do you use for interpolating?

L109-122: A table with the different main thicknesses estimates would be useful and make comparison easier for the reader. I would e.g. include the mean thickness for

each experiment, the mean thickness in the consensus estimate and Bohleber et al, and perhaps the thickness at the borehole locations. I know this is a brief communication and you are not allowed more figures, but maybe as a supplement.

L122: is it possible to calculate an uncertainty on the mean numbers? e.g. by leaving some GPR points out of the simulation and using those points for validation? Or if that would be too much work, you could give an approximation from the core location values (but then only for 2000). You already give it in percentage in the discussion, but here you could use the maximum absolute value.

L124-174: The discussion would benefit from a short discussion on model uncertainties. For example for the constant viscosity runs, did you conduct a sensitivity analysis? Can you give an approximate uncertainty estimate of the SMB? And are there any uncertainties associated with the use of SIA?

L181-183: Why did you use a method which uses the SIA if the glacier is dynamically inactive? Would a plastic approach not be a better choice? Also, I think this section would fit better in the discussion.

L184-190: I was a bit puzzled on how you reach this conclusion. You suddenly mention "mean viscosity" experiments for NIF, although you did not mention this anywhere in the paper (Only for KG, as written in Table 1). For all three experiments, you always generated a viscosity field from observations for NIF (first from the margins, then using Bohleber et al data). You write that "the reconstructions reveal that if there are no thickness observations available, better results can be achieved with a mean viscosity value as input for ice thickness, instead of margin ice thickness generated from DEMs and glacier outline difference" but from what do you reach this conclusion? For KG you wrote the results for the margin method and the viscosity method were almost equal (and you use the margin method to get the mean viscosity in the first place), and for NIF you did not test it. Please clarify. And if you did do the mean viscosity test for NIF too, you should provide it in the paper.

L196-198: Wouldn't how well the margin method → mean viscosity method works depend on the size of the glacier?

3. Technical comments

L10: Add the thickness in 2000 too

L11: Write the unrealistically thick value

L11: change "meanwhile" to "have become"

L13: change "indicator" to "indicators"

L14: delete "As"

L20: delete "to"

L24: "assessment on" to "assessment of"

L25-28: You haven't introduced what you will do in this study yet, so a bit odd to talk about comparison already. I would suggest changing to: "A recent study attempted to reconstruct the distributed ice thickness for all glaciers outside of Antarctica using a consensus of up to 5 models (Farinotti et al. 2019). This estimate generated ice thicknesses estimates for Northern Icefield (NIF) and Kersten Glacier (KG) using ensembles of 2 and 3 models, respectively." Then at the end of line 37 you can add "The resulting thickness estimates are then compared with the consensus estimate" or similar.

L28-31: I would suggest dividing the sentence in two to make it easier to read: "... (Farinotti et al. 2019). In addition, it was recently discovered that KG has separated into two fragments, which is not in agreement with the estimated high thickness values in the study." I would also add a citation for the separation.

L34: I would suggest adding a line describing the model here, e.g. something like L80-83. Currently you mention a SMB model in L 39 without introducing that you even use

it first.

L39: either delete "the distributed surface mass balance (SMB) model and" or introduce the model in the introduction.

L41: define DEM the first time you use it

L41-43: missing reference for SRTM and Landsat 5

L43: change "from a merge of" to "by merging"

L46: reference Fig 1 after describing the redefinition

L46: Future separation? Earlier you wrote it already separated?

L47: delete "apart from" and add "were" before drilled

L48: can you add the borehole locations to figure 1 instead? It would be nice to have all the observations in the same figure.

L48: Definite GPR first time you use it

L54: change "showed a mean" to "had a mean"

L54: give the value for NIF, "similar value" is too vague

L61-65: You should explain the reason for the model changes first, as it will be easier for the reader to follow. E.g. "The full MB model has only previously been verified for KG. However, because of the low slope angles of NIF, meltwater cannot run off from the surface of its planar top before refreezing sets in (Mölg and Hardy 2004), which was not captured by the model. Therefore we upgraded the model so that refreezing of meltwater is allowed on a bare ice surface with a slope angle below 5 degrees. With these changes, the model is capable of reproducing the observed surface height changes observed by a Sonic Ranger mounted to the AWS."

L76: change "nowadays" to "currently" or "2011"

L89: change "increase" to "increased"

L90-91: I suggest changing the structure so the reasoning is before the how, e.g.: "In order to smooth the surface slope during reconstruction we use use the coupling length parameter, which is defined a a multiple of the local ice thickness."

L95: add "by" before "combining"

L98: the values are inferred and then the values are interpolated for the whole area?

L117: change "a distribution" to "the distribution"

L144: reference is missing a year

L147: what is "the better model"?

L149: change the end of the sentence to ".. the consensus estimate underestimates the the thickness at these points."

L165: mention the 10 and 5 m experiments in methods

L169: remove "where the very high . . . as well"

L178: remove "became ice free or"

---

## Author Comment (AC1) · 25 Jun 2020

**Reply to Referee Comments**

First of all, we want to thank the referee for the critical and constructive comments on our manuscript. We considered all comments. Our replies/actions are indented and given in blue font.

Summary

The manuscript presents estimates for the ice thickness distribution for the glaciers on Mount Kilimanjaro. The estimates refer to the years 2000 and 2011, and are based on a combination of in-situ observations, past ice thickness reconstructions derived for areas that are now ice free and a numerical, ice-flux based approach. The paper seems to have two main points. For one, the available global-scale ice thickness estimates seem to have overestimated the ice thickness for one of the two investigated glaciers. For another, the idea of using a combination of past and present digital elevation models (DEMs) to derive ice thickness observations that can be passed to ice-flux estimation approaches is suggested to hold promise for future applications. The paper has a general good quality, and the findings are certainly worth conveying to the larger audience. Slight improvements seem necessary in the way that individual details are presented. The discussion section could benefit from a somewhat more substantial revision.

Major Comments

- Somehow, I was left in doubt on how the available Ground Penetrating Radar (GPR) measurement enter the game. They are briefly mentioned in the Data section (L. 48), do not show up in the Methods, and re-appear in the Discussion (L. 145). In particular, clarification is required for what the mentioned "assimilation" (L. 33 and 145) actually entails. As the manuscript stands now, no information is provided, and that should be rectified.

    We agree that the article format required us to shorten many technical details. The old document did however specify that viscosity values are computed at the location where thickness values are available (L86-87). In section 3.5, we now further expanded on the details of how GPR measurements are used. We hope that these extra sentences provide the necessary clarification.

- I had some reasonably hard time in following the discussion. I found it particularly hard to keep track of the many comparisons done for the two glaciers, targeting at the three Experiments performed in the work itself, the two (or three?) models used in the consensus estimate, and the two available sources of in-situ observations (boreholes and GPR data). To me, it would seem natural to show a figure depicting the various model results along the available GPR transects. Since both surface DEM and thickness are available for any of the various results, all information required to generate such a plot seems available. Most likely, this would help the readers to better grasp the main outcome of the discussion which, as far as I understand, rather focuses on the performance of the consensus estimate than on the results of the manuscript itself?

    The consensus estimate shows a mean of three (Kersten Glacier) and two (Northern Icefield) separate models. We have decided to omit discussing the models in the consensus estimate separately to avoid misunderstandings.
    As you mentioned in the annotations directly in the manuscript, Farinotti et al.'s consensus estimate has a twice as high ice thickness for Kersten Glacier, while it underestimates the thickness at the boreholes, which are located on the Northern Icefield.
    We reworded parts of the discussion to make it easier to understand by clarifying whether the discussion is about the Northern Icefield or Kersten Glacier.
    We chose to not change the figure as showing the thickness along the GPR transects would not adequately depict the ice thickness distribution across the whole Northern Icefield. Thickness

surveys were only available for NIF. These were directly assimilated by our method and are reproduced. The only thing such a profile graph would show that other approaches deviate. We therefore decided to extract the thickness values at the unconsidered ice core locations and added them into Supplementary Table 1.

- An important point of discussion that seems to have been missed is that ice thickness estimation approaches as used in this study require the investigated glaciers to have some ice flux. Otherwise, the main idea behind the approaches somewhat breaks down. This point is skimmed in the Conclusions & Outlook section (L. 183) but would probably deserve some space in the Discussion section as well. May it help to explain some of the discrepancies noted between model results and observations?

   As Kersten Glacier is located on the steep flank of Mt. Kilimanjaro, we expect some glacier deformation with a clear directional preference even if rates remain small. For NIF, we agree with the reviewer that the situation is more complex. Its central areas are characterized by flat plateaus and the abrupt step changes in the topography over the cliff features. As we suspect little deformation, we can only alleviate this concern by pointing to the error assessment in Fürst et al. (2017). The approach has there been applied to an ice-cap geometry on Svalbard. There it is shown that error estimates associated to the thickness reconstruction increase substantially towards the flat interior where no thickness measurements are available. The reason is that the associated error estimates are inversely proportionate to the ice flux. For NIF, we are however in the favorable position that thickness values were measured over the flat plateau area giving some confidence in the results. We inserted a brief discussion of this issue into the discussion section.

- The last few sentences of the Conclusions & Outlook (L. 190-198) seem the paper's strongest and most valuable point. Shouldn't these implications be highlighted in the abstract as well?

   Due to the abstract being limited to a maximum of 100 words, we were unable to highlight it in the abstract as well.

Minor Comments

1) There are several undefined acronyms, including, amongst other, SRTM at L. 35, MB at L. 61, TDX at L. 66.

   We added the definitions for the previously undefined acronyms.

2) I could not follow the logics exposed at L. 61-63. According to the sentence, the surface mass balance model applied in the study was "slightly altered" because (sic) "it was never tested for Kersten Glacier before". I imagine that the model was actually tested by the authors before altering it, and that the matter is only one of wording?

   The surface mass balance model has previously only been testes on Kersten Glacier. After applying the model with the exact same parameters and settings on the Northern Icefield, we found that it could not reproduce the observed surface height changes measured by the Sonic Ranger mounted to the Automatic Weather Station on the flat parts of NIF. We did test different ways within the scope of the model that would influence the model output to better fit the measurements. We believe that in this case it is a matter of the wording used in the manuscript and we adjusted it to reflect that.

3) At L. 69-72 the authors state that they removed all positive elevation differences from the analysis because such positive changes are "unlikely" to happen. The issue is that this removal apparently affects

some 15% of the area of the Northern Icefield, which calls for some more detail. For example: What is the spatial distribution of these removed cells? Is it completely scattered, suggesting random noise, or is it clustered, indicating that the signal might be real after all? What is the confidence in the individual DEMs? Etc.

> The referee rightly asks for more clarification here. In this section, we failed to clarify that this selection only concerns the DHDT values that are later used to determine past thickness observations in the nowadays ice-free areas. We adjusted our explanation accordingly. Positive DHDT values cannot be considered in the reconstruction because they imply that the formerly ice-covered area had a lower elevation than the nowadays ice-free part. As we aim for distilling useful information from the retreat these values could only be ignored.

4) I was not able to follow L. 90-94. A "coupling length parameter" is introduced without further explanation (I assume the definition is found in Fuerst et al. 2017, which is ok) and, as far as I understand the wording, is first said to control how the surface DEM is "imprinted in the thickness field" (I'm not entirely sure what this means) and later said to control the "smoothness" on not further specified "flux streamlines". I don't want to exclude that the wording makes perfect sense to a reader familiar with the details of Fuerst et al (2017) but I think that some additional words of explanation will help the majority of the readership.

> The coupling length parameter is introduced in Fürst et al. 2017 and controls the horizontal smoothing of the surface slope field with the aim to infer smooth streamlines for the flux computations. We reworded the sentence for clarity.

Line-by-line Comments

A (rather long, I apologise) set of line-by-line comments is found in the annotated document, attached to this review. The comments provided above are contained therein as well.

> In our response below, we only address line-by-line comments that were not addressed above and that do not refer to style, punctuation, grammar, etc.

L. 10: Please state at least a standard deviation.

> We have calculated a mean relative (absolute) error of 26% for the reconstructions at the borehole locations. The value is not small as it exceeds error estimates for the majority of glaciers on Svalbard (Fürst et al., 2017). This value can only be a rough orientation for the uncertainties associated with our reconstruction and we therefore refrain from stating it in the abstract. Yet we included it in the results and the conclusions.

L.11: how is it for NIF?

> We added details for NIF.

L. 27: If the "results of this study" are mentioned, shouldn't they be introduced first? At this stage of the text, the reader doesn't really know yet what the study will be about.

> We reworded the sentence and removed "results of this study".

L. 31: From the context, this "there" seems to refer to the dataset of Farinotti et al., not to KG glacier itself (which is what the sentence seems to say). Possibly reword slightly?

> We reworded accordingly.

L. 32: I'm not sure to understand the meaning of "for the first time". The sentence seems to say that the approach existed before but that no thickness measurements were assimilated so far. However, this is probably not how the sentence was meant?

> For the first time referred to the reconstruction approach being used on Mt. Kilimanjaro for the first time.
> We reworded accordingly.

L. 34: What is the meaning of "thickness input" here?

> Thickness input refers to different data sets of ice thickness observations used as input for the reconstruction approach.
> We reworded accordingly.

L. 35: The wording is slightly confusing: it seems to imply that "satellite information" does not qualify as "observational data". Does the "observational data" only refers to "ground-based observational data" then? And why are the thickness observations called "ground truth" in the abstract then?

> "Observational data" refers to measured ice thickness data, including radar measurements (such as the Bohleber et al. GPR data) as well as the ice core measurements (Thompson et al.), but these observational data sets are not available for Kersten Glacier.
> We reworded accordingly.

L. 35: I'm not sure, which one were the first and the second? Is the first one the one introduced with L.32, or are both first and second referring to what follows in L.32-33?

> We reworded the passage for clarity.

L. 35: Consider providing the resolution explicitly. What is "very high"? 10m, 1m, 10cm?

> Very high resolution in this case means 0.5 m ground resolution. We added the information into the text.

L. 39: I'm not following: Which is "THE distributed SMB model"? There was no SMB model mentioned so far, was there?

> We reworded for clarity and added a cross-reference to Section 3.1 in which the SMB model is described.

L. 43: "from a merge" or "by differencing"? I imagine the latter? Otherwise I'm not sure to understand what is happening.

> Two separate TanDEM-X scenes were merged and then by differencing them from the SRMT DEM, the surface height change was generated.
> Reworded for clarity.

L.46: Please point at a figure where this can be seen. As now, the sentence is pretty abstract.

> Added reference to the corresponding figure (Fig. 1).

L.50: linearly interpolating (I imagine?)

> Adjusted phrase accordingly.

L. 54: I'm not sure: "found" by whom? By the Bohleber et al. study? Or by the Farinotti et al. one?

The consensus estimate provided a similar value.
Rephrased the passage accordingly.

L. 63: Can a rational be given for this slope angle threshold? Is the idea that for steep slopes, the meltwater runs away and therefore does not refreezes in place? I'm not entirely sure I would agree with that.

Yes, the reviewer is correct about the basic idea, but not about the fact that meltwater on steep slopes cannot refreeze in the model. The value 5° is an effective compromise to prevent runoff from the almost horizontal surfaces of the Northern Icefield, since there are virtually no surfaces in this portion of the glacier that would be steeper than 5°. Meltwater, however, can still refreeze in the model on steeper surfaces, which is described in one of the model reference papers (Mölg et al., 2009). However, note that the modification only applies to bare ice (the standard code deals with refreezing only in presence of a snow pack).
We added a sentence to clarify the 5° threshold.

L. 73: I don't understand the meaning of "margin" here. A little wordy, but may "Past ice thickness for areas that have become ice free" be an alternative?

We decided to stick with "margin" as this phrase is used throughout the manuscript and is defined in Section 3.3.

L. 85f: a) Please don't mix the notation $^{-1}$ and $/$. b) I believe this $/$ should not be here at all? (See Pattyn's Equation 11); From the equation above, I understand that this was set to n=3?

We have corrected the notation as suggested.

L.86: I'm somewhat guessing but I imagine that, rather being "quantified", $B$ is "tuned" as to ensure that the flu solution matches the ice thickness.

We changed the wording to "tuned" as suggested.

L. 92: I'm not sure to understand what this means.

The "step in the elevation profile over ice cliffs" refers to the "steep elevation increase at the vertical ice cliffs".
Reworded accordingly.

L.98f: I'm not sure: What was done in Experiment 1 then? I understood that this averaging happened in that experiment already? If not, what viscosity value were used for the locations at which there were no ice thickness observations?

In Experiment 1 the generated margin thicknesses are used as thickness observations. During this experiment, the mean viscosity is generated within the reconstruction approach. This mean viscosity is in turn used in Experiment 2 as thickness input.
Rephrased for clarity.

L. 100: What is the meaning of "generic data" here?

Generic data refers to the margin thickness data.
Reworded for clarity.

L. 109: As far as I'm concerned, this sentence an be removed.

We decided to remove the sentence.

L. 124: Please clarify: is this wording referring to the results of Farinotti et al.?

> Yes, this phrase refers to results from Farinotti et al.
> Added source for clarity.

L. 127ff: I'm not sure to follow, is this discussion still referring to the "consensus thickness map"? Somehow, the focus seems to have shifted without noticing; Now I'm lost: What is this "second run" referring to? Is this Experiment 2? That's what the caption of Fig. 2 suggests. The wording is confusing.

> Added Experiment numbers for clarity.

L. 132ff: Please split this sentence in at least two parts. I apologize, but I could not follow.

> Split the sentence for easier readability.

L. 145: What is the meaning of "assimilated" here? Was the viscosity tuned again, as it was done for the ice thickness at the margin?

> For NIF, the GPR measurements are used as thickness input. This was referred to here.
> Rephrased for clarity.

L. 150: The concept of an "error margin" was not introduced, was it? I'm not sure to understand what is meant by that.

> Reworded for clarity.

L. 151: I'm again in the need of guessing: are these "separate entities" something defined by the RGI? Fig. 1 doesn't show three entities on NIF, though?

> The three different glacier entities are defined by the RGI and are also used in the Farinotti et al. consensus estimate. We have merged the three entities into one for our reconstruction, as the approach by Fürst et al. assigns the glacier margin an ice thickness of 0, which did not appear reasonable for the boundary lines between the different entities on the Northern Icefield.

L. 151ff: Sorry, I'm lost here: what is "model 1"? Is this meant to refer to Experiment 1 perhaps? This would be my first guess, but the next sentence is somewhat at odds with that. Has it something to do with "model 01" mentioned at L. 155?

> Model 1 (and later Model 01) refers to one of the models from the consensus estimate.
> Removed the single model data for more clarity and focused only on the consensus.

L. 159: Sorry, I'm lost again: didn't L. 155 say that the consensus has two models? Where is the third one coming from now, or why wasn't it mentioned at L. 155?

> The consensus estimate is made up of two model for NIF and three models for KG.
> Rephrased for clarity.

L. 166ff: Is my understanding correct: For NIF the consensus thickness is thus relatively close to both the GPR measurements and the results presented in this paper? This should probably be said explicitly as well, I imagine?

> Yes, the mean ice thicknesses for the consensus estimate, our Experiments 2 and 3, as well as the reconstructions by Bohleber et al. are relatively close to one another. We have added a sentence stating this into the conclusion.

L. 176: Well, why is the volume never mentioned in the text then?

We replaced "volume" with thickness.

L. 181: What is the meaning of "retreat information"?

This refers to the lateral glacier retreat information, which was digitized from Landsat scenes and used in the margin thickness generation (Section 3.5)

L. 184f: Wait, didn't this "mean viscosity" come from the "margin ice thickness generated from DEMs and glacier outline" as well? How can this claim be made then?

The mean viscosity is generated from the margin ice thicknesses, but while local uncertainties from the margin thickness generation can influence the ice thickness distribution over the whole glacier (Fig. 2A KG), the mean viscosity shows a smoother ice thickness distribution, which seems more likely for Kersten Glacier (Fig. 2B KG). But as there are no thickness observations available for KG we cannot verify if the smoothed ice thickness distribution (Fig. 2B) is closer to reality or not.

L. 189f: I might be completely off track, but where would this mean viscosity come from at this stage? And isn't this claim somewhat in contradiction with what said at L. 135-136, i.e. that "the use of margin thickness information, generated from outline differences enabled a local glacier-specific viscosity tuning which might be preferential to an empirical temperature relation" (since, I assume, the latter would result in a mean viscosity as mentioned in the sentence)?

The mean viscosity is generated from within the thickness reconstruction approach. It is generated during the reconstruction using the margin thickness information generated from glacier outline differences. This means that by using glacier outline differences we can generate margin thickness information and then in a second step the mean viscosity. The results from our experiments show that for KG, where no ground/radar thickness observations were available, using the mean viscosity creates a smoother ice thickness distribution. This result might then be used preferential to approaches using empirical temperature relations to assess a glacier ice thickness as it is locally tuned from the direct glacier retreat as seen in satellite data.

---

## Author Comment (AC2) · 25 Jun 2020

**Reply to Referee Comments**

First of all, we want to thank you for the critical and constructive comments on our manuscript. We considered all comments. Our replies/actions are indented and given in blue font.

General Comments

In this study, the authors estimate the ice thickness of Northern Icefield and Kersten Glacier on Mt. Kilimanjaro in 2000 and 2011 using the ice thickness approach by Fürst et al (2017). Three different "experiments" are conducted to estimate the ice thickness, which either improves or are within the estimates from previous studies. The study makes good use of the few available observations, and the method and results are generally sound and interesting. I know this is a brief communication, but there are some key pieces of information that are missing within the data and model descriptions which I think are necessary to understand the manuscript, and the results would benefit from a discussion of uncertainties. In addition, the conclusion needs to be rewritten, as it does not seem to fit with the rest of the paper. In general, the manuscript would also benefit from an increase in specificity and clarity, as I often had trouble following the text. I hope the technical comments are useful for improving this.

Specific Comments

L40: Location of the AWS is not on Figure 1. Also, for how long a period has the AWS measured and what components are measures (and with what uncertainties)?

> The AWS collected data from February 2005 to September 2013. It is located on Kersten Glacier at 5873 m.a.s.l. and the measurements include incoming and outgoing radiative fluxes (longwave and shortwave) with an accuracy of ±10%, air temperature (±0.2°C), relative humidity (±2% units), wind speed and wind direction (±0.3 ms$^{-1}$ and ±5°), air pressure (±0.2hPa) and the distance to the surface (±0.4%) (Section 3 c. in Mölg et al. 2009a).

> Due to the limited space in a "brief communication" we added a reference to the corresponding article by Mölg et al.

L48-49: This is not quite correct. The thickness estimate was created from the GPR by doing kringing interpolation, and it is an estimate from the whole area, not just the flat central part. Later in the manuscript (L 169), you seem to use the estimate that Bohleber created from the DEM, so I would mention that result here too. E.g. "In addition, ground penetrating radar (GPR) profiles from September 2015 (Fig. 1) were created by Bohleber et al. (2017). Using a kringing interpolation and the KILISoSDEM, the authors estimated the mean thickness to be between 21.2 ± 1 m and 27 ± 2 m."

> Implemented your suggestion into the manuscript.

L55-56: The Bohleber estimate is from 2015 and the consensus estimate is from 2000, so that should also contribute to the differences.

> Added a sentence concerning the different years for clarity.

L39-56: For most of the observations you do not provide uncertainty estimates

> We deliberately decided to not account for the input uncertainties in this case study as the focus is to exploit multi-temporal satellite information to better constrain a thickness reconstruction. Uncertainty consideration are covered in the methodological study by Fürst et al. (2017) comprising a spectrum of leave-out and sensitivity experiments. In light of the short communication format, it appears distracting to expand on the propagation of the input uncertainties into the final result. Moreover, input fields (SMB, DEM, outlines, thickness measurements) are often not necessarily provided with a robust error estimate. We therefore decided to refer the interested reader to Fürst et al. (2017) concerning the associated uncertainties.

L61: You have a point measurement in one location. How to you get distributed mass balance maps from one point on one glacier? And do you use the mean SMB from 2005-2013 for the 2000 and 2011 estimation? If you do, you should mention this as a possible source of uncertainty (and if you don't, how do you find the 2000 SMB?).

> The surface mass balance model (Mölg et al. 2008, 2009a) creates the distributed mass balance based on a DEM (we used the SRTM), which creates the lower boundary conditions for the model and the meteorological data from the AWS, which are used as the model driver (Mölg et al. 2009a). We use the mean SMB for both, the 2000 and 2011 glacier states.

L61-65: Again, what do you use as forcing for NIF if the AWS is on KG? And how do you use the sonic ranger to test refreezing on NIF if it is mounted on KG? In addition, you should mention the sonic ranger in the data section and not only in the methods.

> We use the meteorological data gathered by the AWS on KG as forcing on NIF as well. There is an WS installed on NIF from which we use a plotted time series of the sonic ranger measurements to which we compare our modelled accumulated surface height change. As the ice thickness reconstruction use the mean annual surface mass balance as input, we mainly used the total accumulated surface height change over the time period/ at the end of the modelling period (2013 September) to compare our results to. So the climatic variables (T, RH, ..) are the same for NIF and KG, but the topographic/elevation data differs, as this is directly calculated from the digital elevation model SRTM.

L87: What method do you use for interpolating?

> The method used for interpolation is Natural Neighbor/Sibsonian Interpolation.

> We mentioned the method now in the manuscript.

L109-122: A table with the different main thicknesses estimates would be useful and make comparison easier for the reader. I would e.g. include the mean thickness for each experiment, the mean thickness in the consensus estimate and Bohleber et al, and perhaps the thickness at the borehole locations. I know this is a brief communication and you are not allowed more figures, but maybe as a supplement.

We added a table containing the mean thickness estimates for NIF and KG and the thickness at the Thompson et al. (2002) borehole locations to the supplement (Supplementary Table 1).

L122: is it possible to calculate an uncertainty on the mean numbers? e.g. by leaving some GPR points out of the simulation and using those points for validation? Or if that would be too much work, you could give an approximation from the core location values (but then only for 2000). You already give it in percentage in the discussion, but here you could use the maximum absolute value.

We added the suggested approximation of absolute values at the core locations. For Experiment 1(2) the ice thickness at the core locations differ by 19.9 m (4.4 m) at C1, 23.9 m (8.3 m) at C2 and 36.6 m (26.1 m) at C3.

L124-174: The discussion would benefit from a short discussion on model uncertainties. For example for the constant viscosity runs, did you conduct a sensitivity analysis? Can you give an approximate uncertainty estimate of the SMB? And are there any uncertainties associated with the use of SIA?

The inferred **viscosity values** not only depend on the structural and temperature properties of the glacier body. They are also affected by the uncertainties of all other input fields and measurements. As the input uncertainty is already analyzed in depth by withholding GPR measurements in Fürst et al. (2017), it seemed redundant to repeat this exercise here. Certainly, in light of the short article format.

We cannot give an approximation on the **SMB uncertainty**, but as previously shown in Fürst et al. (2017) its influence on the ice thickness reconstruction is only minor. It was shown that, by changing the SMB input drastically, the mean ice thickness is reduced by 5% and the estimated ice volume by 4%. These values were found when the least amount of direct thickness measurements was assimilated. Moreover, this influence was estimated for various glacier geometries, including an ice cap, on Svalbard. They also noted that, where ice thickness data is available, the influence of SMB input is compensated by direct observations (Fürst et al. 2017).

First of all, the **SIA** is a key component of this type of reconstruction. An expansion to include the solution of more complete forms of the force balance would require fundamental adjustments in the method. Though more complete, the problem might become even less well-posed and the computing requirements would increase unproportionate. Some of the uncertainties associated to the choice of the SIA are covered in Fürst et al. (2017).

L181-183: Why did you use a method which uses the SIA if the glacier is dynamically inactive? Would a plastic approach not be a better choice? Also, I think this section would fit better in the discussion.

We use the SIA in our reconstruction, as it is the method implemented in our reconstruction approach. KG is located on the steep flank of Mt. Kilimanjaro and we expect some ice motion. For NIF, this issue might be more relevant, and we expanded the discussion of this aspect in the revised manuscript. In such setups, the mass-conserving SIA approach is not ideal. We have no model to use a plastic flow assumption, so this approach was also not viable for us.

Concerning a plastic approach, it would certainly be an alternative here. Yet such approaches have often been applied in flowline setups with appropriate spatial averaging of the geometric input. Although one could theoretically apply them in 2D to each grid point, it would require an extra article to assess what the best strategies would be for spatial smoothing of the required input. We are unaware of a precursor study that applies the perfect plasticity concept in 2D

(without final spatial interpolation) that is readily transferable to the complex topography of NIF.

L184-190: I was a bit puzzled on how you reach this conclusion. You suddenly mention "mean viscosity" experiments for NIF, although you did not mention this anywhere in the paper (Only for KG, as written in Table 1). For all three experiments, you always generated a viscosity field from observations for NIF (first from the margins, then using Bohleber et al data). You write that "the reconstructions reveal that if there are no thickness observations available, better results can be achieved with a mean viscosity value as input for ice thickness, instead of margin ice thickness generated from DEMs and glacier outline difference" but from what do you reach this conclusion? For KG you wrote the results for the margin method and the viscosity method were almost equal (and you use the margin method to get the mean viscosity in the first place), and for NIF you did not test it. Please clarify. And if you did do the mean viscosity test for NIF too, you should provide it in the paper.

> We fear that we have not been careful enough in presenting the experiments which raised this concern. In the case of 'directly using lateral thickness information' and in the case of the 'mean viscosity', the thickness information from the retreat (ice-free area) is used. The difference is only how the reconstruction deals with this data. The two options are that the viscosity of each 'lateral thickness point' is used individually for an interpolation over the domain, resulting in a spatially variable ice viscosity (Experiment 1, NIF and KG). Otherwise, the viscosity point information is simply averaged, and a uniform value is used for the entire glacier (Experiment 2 and 3, KG).

L196-198: Wouldn't how well the margin method / mean viscosity method works depend on the size of the glacier?

> We believe that the size of the glacier would most likely influence the outcome of the margin/mean viscosity method, but we have not tested the approach on glaciers of different sizes so we cannot comment on that further. The Kilimanjaro setup is quite special, and it is difficult to assess the glacier size dependence. Yet, glacier retreat is mostly expressed at low elevations. It is there that we expect to acquire past thickness values from multi-temporal satellite information. As the frontal area represents an increasingly smaller portion of the entire system as glaciers become larger, the size dependence is certainly an interesting question. We can unfortunately not answer this here on the basis of the two very different glacier types on Mt. Kilimanjaro.

Technical Comments

L10: Add the thickness in 2000 too

> We refrained from adding the 2000 thicknesses into the abstract as the word limit did not allow us to explain the difference (thickness increase) between the 2000 and 2011 reconstructions sufficiently and we believe it might cause confusion without the proper explanation.

L11: Write the unrealistically thick value

> Changed the manuscript accordingly.

L11: change "meanwhile" to "have become"

    *Changed the manuscript accordingly.*

L13: change "indicator" to "indicators"

    *Changed the manuscript accordingly.*

L14: delete "As"

    *We decided to stick with this wording.*

L20: delete "to"

    *Changed the manuscript accordingly.*

L24: "assessment on" to "assessment of"

    *Changed the manuscript accordingly.*

L25-28: You haven't introduced what you will do in this study yet, so a bit odd to talk about comparison already. I would suggest changing to: "A recent study attempted to reconstruct the distributed ice thickness for all glaciers outside of Antarctica using a consensus of up to 5 models (Farinotti et al. 2019). This estimate generated ice thicknesses estimates for Northern Icefield (NIF) and Kersten Glacier (KG) using ensembles of 2 and 3 models, respectively." Then at the end of line 37 you can add "The resulting thickness estimates are then compared with the consensus estimate" or similar.

    *Reworded the passage according to the suggestion.*

L28-31: I would suggest dividing the sentence in two to make it easier to read: ". . . (Farinotti et al. 2019). In addition, it was recently discovered that KG has separated into two fragments, which is not in agreement with the estimated high thickness values in the study." I would also add a citation for the separation.

    *Divided sentence as suggested. Added reference to the Landsat scene used in the study.*

L34: I would suggest adding a line describing the model here, e.g. something like L80-83. Currently you mention a SMB model in L 39 without introducing that you even use it first.

    *Added information on the SMB model in the introduction. As this manuscript is a "brief communication" we refrained from adding further information on the reconstruction approach in the introduction. We added a cross-reference to the corresponding section 3.4.*

L39: either delete "the distributed surface mass balance (SMB) model and" or introduce the model in the introduction.

    *We briefly introduced the model in the introduction.*

L41: define DEM the first time you use it

 Added definition of DEM.

L41-43: missing reference for SRTM and Landsat 5

 We added references to the data sets used.

L43: change "from a merge of" to "by merging"

 Reworded to "by differencing from a merge of two …"

L46: reference Fig 1 after describing the redefinition

 Added reference to Figure 1.

L46: Future separation? Earlier you wrote it already separated?

 We anticipate a future separation of the Northern Icefield. Kersten Glacier has already separated.
 Reworded for clarification.

L47: delete "apart from" and add "were" before drilled

 Deleted words as suggested.

L48: can you add the borehole locations to figure 1 instead? It would be nice to have all the observations in the same figure.

 Added borehole locations to Fig. 1 and removed them from Fig. 2.

L48: Definite GPR first time you use it

 Defined acronym.

L54: change "showed a mean" to "had a mean"

 Changed wording as suggested.

L54: give the value for NIF, "similar value" is too vague

 Removed passage from manuscript.

L61-65: You should explain the reason for the model changes first, as it will be easier for the reader to follow. E.g. "The full MB model has only previously been verified for KG. However, because of the low slope angles of NIF, meltwater cannot run off from the surface of its planar top before refreezing sets in (Mölg and Hardy 2004), which was not captured by the model. Therefore we upgraded the model so that refreezing of meltwater is allowed on a bare ice surface with a slope angle below 5 degrees. With

these changes, the model is capable of reproducing the observed surface height changes observed by a Sonic Ranger mounted to the AWS."

*Rephrased the section for clarity with the suggestions in mind.*

L76: change "nowadays" to "currently" or "2011"

*Changed "nowadays" to "currently".*

L89: change "increase" to "increased"

*Changed word as suggested.*

L90-91: I suggest changing the structure so the reasoning is before the how, e.g.: "In order to smooth the surface slope during reconstruction we use use the coupling length parameter, which is defined a a multiple of the local ice thickness."

*Changed wording as suggested.*

L95: add "by" before "combining"

*Added the word "by" as suggested.*

L98: the values are inferred and then the values are interpolated for the whole area?

*We rephrase this passage and hope that it became clearer now.*

L117: change "a distribution" to "the distribution"

*Reworded the sentence.*

L144: reference is missing a year

*Added missing year to the reference. The reference is Thompson et al. 2002.*

L147: what is "the better model"?

*Removed the distinction of the two models that make up the consensus estimate for NIF for easier understanding and reworded the passage.*

L149: change the end of the sentence to ".. the consensus estimate underestimates the the thickness at these points."

*We rephrased a large part of the discussion for clarity, so the sentence referred to here was completely changed.*

L165: mention the 10 and 5 m experiments in methods

Mentioned the 10 and 5 m experiments in the methods section 3.4.

"With the higher DEM quality in 2011, the resolution was iteratively increased from 25, via 10 and 5, to 2 m."

L169: remove "where the very high . . . as well"

We rephrased a large part of the discussion for clarity, so the sentence referred to here was completely changed.

L178: remove "became ice free or"

Rephrased the sentence to "in areas that became ice-free in the last decade."

---

## Author Response (AR3)

**Editor Decision: Publish subject to minor revisions (review by editor)**

Dear Editor,

We want to thank you again for all the critical and constructive comments on our manuscript. We have carefully considered your comments. Our point-by-point response to the reviewers' comments follows below. Our replies/actions are indented and given in blue font.
* * *
Line 11: "...and that allow glacier-specific calibration". Unclear what "that" refers to.

> In this case "that" refers to the areas that have become ice free. Changed that to "therefore" for clarity.

Line 32: Change "illustrate" to for example "found"

> Changed wording as suggested.

Lines 42-44: I suggest changing to: "The change in surface height between 2000 and 2011 was found by differencing a merge of two TanDEM-X radar images (28 January 2011, 4 April 2011) and the SRTM DEM."

> Changed the sentence as suggested.

Line 57: Strictly speaking, isn't this an extrapolation? The surface elevation change is extrapolated to the point in time when the GPR thicknesses were acquired.

> Yes, the editor is correct. We changed "interpolating" to "extrapolating" in the manuscript.

Lines 58-65: I am not sure that this (very useful) paragraph belongs in the Data section. Consider if this would be a better fit in Section 3 (for example under the description of the experimental set up) or in the Discussion section.

> Moved the paragraph to Section 3.3 Ice thickness reconstruction.

Line 71: change: "...being driven by..." -> "...driven by..."

> Removed "being".

Line 73: delete "observed"

> Deleted "observed".

Section 3.2: This section is now very short and one of the lines is basically repeating what has already been mentioned in Section 2. Consider merging this section with the data description, since the manuscript does not deal with the details of TanDEM-X processing anyway.

> Merged this section with the data description.

Line 135: It is good to remind the reader that Experiment 3 is in a different year than Experiment 1+2. For example, the sentence could say "In Experiment 3 (Fig. 1) we now move forward in time to 2011. Here, KG is split into two parts... "

> Adjusted sentence as suggested.

Line 141: Change "orientation" to "guideline" otherwise a reader might think that it is the orientation of the glacier.

Changed wording as suggested.

Lines 154-155: Does the consensus map use a constant viscosity? If yes, is that the explanation for the similarity?

The thickness distribution of Experiment 2 is similar to that of the consensus estimate, but it is difficult to explain solely via the viscosity parameter, as the models included in the consensus estimate show non-negligible methodological differences.

Line 161: Which Experiment is being discussed? Exp 1?

Yes, this sentence refers to Experiment 1. Added reference to the text.

Lines 164-165: "Increased mismatch values, especially for borehole C3, might as well be explained by the very flat plateau." What does this sentence refer to? It implies a change between two values but the mismatches decrease from Experiment 1 to 2. Please clarify. Should "Increased" be "Large"?

"Increased" refers to "large" in this case.

Lines 171-173: This sentence: "Despite that no GPR measurements were considered on Mt. Kilimanjaro, the complex topography posed a similar challenge for the models participating in the consensus." combines two seemingly unrelated sentences. It is not clear how the absence/presence of GPR measurements is coupled to the topography. Please clarify.

Changed the sentence to "The complex topography posed a similar challenge for the models participating in the consensus, especially because no thickness observations were considered for Mt. Kilimanjaro."

Line 180: Please mention the mean ice thicknesses found in Experiment 2: "..increased in comparison to Experiment 2 from XXm to 9.3 m and from 23.4m to 26.6 m, respectively."

Added the mean ice thicknesses from Experiment 2 as suggested.

Lines 181-182: "Resolution can be excluded from a 25 m reconstruction in 2011 (not shown)." - > "Resolution can be excluded based on results from a 25 m reconstruction in 2011 (not shown)."

Adjusted sentence as suggested.

Lines 182-185: "Concerning the 2011 outlines, some internal ice-free areas (on both NIF and KG), present in the RGI, could not be confirmed from the coarse Landsat imagery, resulting in thicker ice. The quality difference between SRTM and KILISoSDEM is certainly also a contributing factor explaining part of the larger thickness values. "
This part is really crucial in order to understand why there is a difference between the 2000 and 2011 reconstructions. The present version leaves several questions unanswered. Firstly, are the ice-free areas real? If yes, is the difference then due to the (incorrect?) outline from the Landsat imagery? Wouldn't you then have gotten better results if you had incorporated the RGI ice-free areas in your 2011 outline? Secondly, what is meant by quality difference? Quality in what? How do the DEMs differ and how does their difference impact your results?

> The ice-free areas on KG and NIF are located very close to the glacier margin, or in areas that have become ice-free by 2011, which means these areas are indirectly accounted for in the reconstruction.
> We did however notice ice-free areas mapped in Bohleber et al. (2017). Bohleber et al. digitized their outline and the ice-free areas from a GeoEye-1 satellite image from 2012, which has a higher resolution than the Landsat image we used so we were unable to detect these smaller ice-free spots.
> The quality difference between the two DEMs is largely based on different ground resolutions. The SRTM DEM is a global dataset with a 30 m ground resolution, whereas the KILISoSDEM only covers the Kilimanjaro area, but has a 0.5 m ground resolution. With a higher DEM resolution, we can also run the ice thickness reconstruction at a higher resolution, which is especially important to accurately reconstruct areas such as the vertical ice cliffs.
> Changed "The quality difference…" to "The difference in resolution…" .

Lines 196-197: The term "data assimilation" is not used in the methods description at all. Considering that the reviewers also asked for more details on this, I ask that you explicitly mention this method, for example, in Section 3.5, e.g., "The method of Fürst et al. uses data assimilation to...". Alternatively, rephrase this part in the conclusion.

> We added a brief explanation for the term "data assimilation" to Section 3.3.

Lines 210-211: "We therefore speculate that thickness information from retreat is most useful in areas that have been dynamically more active in the past." This is a very important point and I think it should also be directly mentioned in the Discussion. As it is now it is buried.

> We briefly mentioned it in the Discussion Section as well (l. 184f. in the tracked-changes manuscript).

Section 6: This section could do with some streamlining and tidying up. At the moment, report on results is mixed with discussions of the utility of retreat observations and ice thickness measurements. I am also confused by the word "assess" - the new ice thickness map is compared to the consensus map but since the latter is likely wrong it seems counterintuitive to assess with consensus map?
I suggest restructuring the section:
Firstly, report on the findings. Mean ice thicknesses for 2000 and 2011 and how the results compare to the consensus map.
Secondly, the use of observations of glacier retreat to constrain ice thicknesses. From your results it would seem that direct ice thickness measurements are most valuable, then glacier retreat observations and then (in the absence of any data) the consensus model approach.
Thirdly, outlook: In the future multi-temporal satellite information might aid in reconstructing glacier thicknesses with the caveat that complex topography (or other factors) might complicate matters.

> We restructured the conclusion as per your suggestions. We assess how our reconstruction is performing in comparison to the current best estimate, which is the consensus estimate.

Fig. 2: I don't see any coloured dots with a magenta outline? I can see magenta contours. Please revise figure caption.

> Adjusted the figure caption.

Dear Editor,

We want to thank you and the two anonymous referees for the critical and constructive comments on our manuscript. We have carefully considered the reviewers' and your comments. Our point-by-point response to the reviewers' comments follows below. Our replies/actions are indented and given in blue font.
* * *
**Reply to Reviewer RC1**

Summary

The manuscript presents estimates for the ice thickness distribution for the glaciers on Mount Kilimanjaro. The estimates refer to the years 2000 and 2011, and are based on a combination of in-situ observations, past ice thickness reconstructions derived for areas that are now ice free and a numerical, ice-flux based approach. The paper seems to have two main points. For one, the available global-scale ice thickness estimates seem to have overestimated the ice thickness for one of the two investigated glaciers. For another, the idea of using a combination of past and present digital elevation models (DEMs) to derive ice thickness observations that can be passed to ice-flux estimation approaches is suggested to hold promise for future applications. The paper has a general good quality, and the findings are certainly worth conveying to the larger audience. Slight improvements seem necessary in the way that individual details are presented. The discussion section could benefit from a somewhat more substantial revision.

Major Comments

- Somehow, I was left in doubt on how the available Ground Penetrating Radar (GPR) measurement enter the game. They are briefly mentioned in the Data section (L. 48), do not show up in the Methods, and re-appear in the Discussion (L. 145). In particular, clarification is required for what the mentioned "assimilation" (L. 33 and 145) actually entails. As the manuscript stands now, no information is provided, and that should be rectified.

  We agree that the article format required us to shorten many technical details. The old document did however specify that viscosity values are computed at the location where thickness values are available (L86-87). In section 3.5, we now further expanded on the details of how GPR measurements are used. We hope that these extra sentences provide the necessary clarification.

- I had some reasonably hard time in following the discussion. I found it particularly hard to keep track of the many comparisons done for the two glaciers, targeting at the three Experiments performed in the work itself, the two (or three?) models used in the consensus estimate, and the two available sources of in-situ observations (boreholes and GPR data). To me, it would seem natural to show a figure depicting the various model results along the available GPR transects. Since both surface DEM and thickness are available for any of the various results, all information required to generate such a plot seems available. Most likely, this would help the readers to better grasp the main outcome of the discussion which, as far as I understand, rather focuses on the performance of the consensus estimate than on the results of the manuscript itself?

  The consensus estimate shows a mean of three (Kersten Glacier) and two (Northern Icefield) separate models. We have decided to omit discussing the models in the consensus estimate separately to avoid misunderstandings.

As you mentioned in the annotations directly in the manuscript, Farinotti et al.'s consensus estimate has a twice as high ice thickness for Kersten Glacier, while it underestimates the thickness at the boreholes, which are located on the Northern Icefield.

We reworded parts of the discussion to make it easier to understand by clarifying whether the discussion is about the Northern Icefield or Kersten Glacier.

We chose to not change the figure as showing the thickness along the GPR transects would not adequately depict the ice thickness distribution across the whole Northern Icefield. Thickness surveys were only available for NIF. These were directly assimilated by our method and are reproduced. The only thing such a profile graph would show that other approaches deviate. We therefore decided to extract the thickness values at the unconsidered ice core locations and added them into Supplementary Table 1.

- An important point of discussion that seems to have been missed is that ice thickness estimation approaches as used in this study require the investigated glaciers to have some ice flux. Otherwise, the main idea behind the approaches somewhat breaks down. This point is skimmed in the Conclusions & Outlook section (L. 183) but would probably deserve some space in the Discussion section as well. May it help to explain some of the discrepancies noted between model results and observations?

As Kersten Glacier is located on the steep flank of Mt. Kilimanjaro, we expect some glacier deformation with a clear directional preference even if rates remain small. For NIF, we agree with the reviewer that the situation is more complex. Its central areas are characterized by flat plateaus and the abrupt step changes in the topography over the cliff features. As we suspect little deformation, we can only alleviate this concern by pointing to the error assessment in Fürst et al. (2017). The approach has there been applied to an ice-cap geometry on Svalbard. There it is shown that error estimates associated to the thickness reconstruction increase substantially towards the flat interior where no thickness measurements are available. The reason is that the associated error estimates are inversely proportionate to the ice flux. For NIF, we are however in the favorable position that thickness values were measured over the flat plateau area giving some confidence in the results. We inserted a brief discussion of this issue into the discussion section.

- The last few sentences of the Conclusions & Outlook (L. 190-198) seem the paper's strongest and most valuable point. Shouldn't these implications be highlighted in the abstract as well?

Due to the abstract being limited to a maximum of 100 words, we were unable to highlight it in the abstract as well.

Minor Comments

1) There are several undefined acronyms, including, amongst other, SRTM at L. 35, MB at L. 61, TDX at L. 66.

We added the definitions for the previously undefined acronyms.

2) I could not follow the logics exposed at L. 61-63. According to the sentence, the surface mass balance model applied in the study was "slightly altered" because (sic) "it was never tested for Kersten Glacier before". I imagine that the model was actually tested by the authors before altering it, and that the matter is only one of wording?

The surface mass balance model has previously only been testes on Kersten Glacier. After applying the model with the exact same parameters and settings on the Northern Icefield, we found that it could not reproduce the observed surface height changes measured by the Sonic

Ranger mounted to the Automatic Weather Station on the flat parts of NIF. We did test different ways within the scope of the model that would influence the model output to better fit the measurements. We believe that in this case it is a matter of the wording used in the manuscript and we adjusted it to reflect that.

3) At L. 69-72 the authors state that they removed all positive elevation differences from the analysis because such positive changes are "unlikely" to happen. The issue is that this removal apparently affects some 15% of the area of the Northern Icefield, which calls for some more detail. For example: What is the spatial distribution of these removed cells? Is it completely scattered, suggesting random noise, or is it clustered, indicating that the signal might be real after all? What is the confidence in the individual DEMs? Etc.

The referee rightly asks for more clarification here. In this section, we failed to clarify that this selection only concerns the DHDT values that are later used to determine past thickness observations in the nowadays ice-free areas. We adjusted our explanation accordingly. Positive DHDT values cannot be considered in the reconstruction because they imply that the formerly ice-covered area had a lower elevation than the nowadays ice-free part. As we aim for distilling useful information from the retreat these values could only be ignored.

4) I was not able to follow L. 90-94. A "coupling length parameter" is introduced without further explanation (I assume the definition is found in Fuerst et al. 2017, which is ok) and, as far as I understand the wording, is first said to control how the surface DEM is "imprinted in the thickness field" (I'm not entirely sure what this means) and later said to control the "smoothness" on not further specified "flux streamlines". I don't want to exclude that the wording makes perfect sense to a reader familiar with the details of Fuerst et al (2017) but I think that some additional words of explanation will help the majority of the readership.

The coupling length parameter is introduced in Fürst et al. 2017 and controls the horizontal smoothing of the surface slope field with the aim to infer smooth streamlines for the flux computations. We reworded the sentence for clarity.

Line-by-line Comments

A (rather long, I apologise) set of line-by-line comments is found in the annotated document, attached to this review. The comments provided above are contained therein as well.

In our response below, we only address line-by-line comments that were not addressed above and that do not refer to style, punctuation, grammar, etc.

L. 10: Please state at least a standard deviation.

We have calculated a mean relative (absolute) error of 26% for the reconstructions at the borehole locations. The value is not small as it exceeds error estimates for the majority of glaciers on Svalbard (Fürst et al., 2017). This value can only be a rough orientation for the uncertainties associated with our reconstruction and we therefore refrain from stating it in the abstract. Yet we included it in the results and the conclusions.

L.11: how is it for NIF?

We added details for NIF.

L. 27: If the "results of this study" are mentioned, shouldn't they be introduced first? At this stage of the text, the reader doesn't really know yet what the study will be about.

We reworded the sentence and removed "results of this study".

L. 31: From the context, this "there" seems to refer to the dataset of Farinotti et al., not to KG glacier itself (which is what the sentence seems to say). Possibly reword slightly?

> We reworded accordingly.

L. 32: I'm not sure to understand the meaning of "for the first time". The sentence seems to say that the approach existed before but that no thickness measurements were assimilated so far. However, this is probably not how the sentence was meant?

> For the first time referred to the reconstruction approach being used on Mt. Kilimanjaro for the first time.
> We reworded accordingly.

L. 34: What is the meaning of "thickness input" here?

> Thickness input refers to different data sets of ice thickness observations used as input for the reconstruction approach.
> We reworded accordingly.

L. 35: The wording is slightly confusing: it seems to imply that "satellite information" does not qualify as "observational data". Does the "observational data" only refers to "ground-based observational data" then? And why are the thickness observations called "ground truth" in the abstract then?

> "Observational data" refers to measured ice thickness data, including radar measurements (such as the Bohleber et al. GPR data) as well as the ice core measurements (Thompson et al.), but these observational data sets are not available for Kersten Glacier.
> We reworded accordingly.

L. 35: I'm not sure, which one were the first and the second? Is the first one the one introduced with L.32, or are both first and second referring to what follows in L.32-33?

> We reworded the passage for clarity.

L. 35: Consider providing the resolution explicitly. What is "very high"? 10m, 1m, 10cm?

> Very high resolution in this case means 0.5 m ground resolution. We added the information into the text.

L. 39: I'm not following: Which is "THE distributed SMB model"? There was no SMB model mentioned so far, was there?

> We reworded for clarity and added a cross-reference to Section 3.1 in which the SMB model is described.

L. 43: "from a merge" or "by differencing"? I imagine the latter? Otherwise I'm not sure to understand what is happening.

> Two separate TanDEM-X scenes were merged and then by differencing them from the SRMT DEM, the surface height change was generated.
> Reworded for clarity.

L.46: Please point at a figure where this can be seen. As now, the sentence is pretty abstract.

> Added reference to the corresponding figure (Fig. 1).

L.50: linearly interpolating (I imagine?)

> Adjusted phrase accordingly.

L. 54: I'm not sure: "found" by whom? By the Bohleber et al. study? Or by the Farinotti et al. one?

> The consensus estimate provided a similar value.
> Rephrased the passage accordingly.

L. 63: Can a rational be given for this slope angle threshold? Is the idea that for steep slopes, the meltwater runs away and therefore does not refreezes in place? I'm not entirely sure I would agree with that.

> Yes, the reviewer is correct about the basic idea, but not about the fact that meltwater on steep slopes cannot refreeze in the model. The value 5° is an effective compromise to prevent runoff from the almost horizontal surfaces of the Northern Icefield, since there are virtually no surfaces in this portion of the glacier that would be steeper than 5°. Meltwater, however, can still refreeze in the model on steeper surfaces, which is described in one of the model reference papers (Mölg et al., 2009). However, note that the modification only applies to bare ice (the standard code deals with refreezing only in presence of a snow pack).
> We added a sentence to clarify the 5° threshold.

L. 73: I don't understand the meaning of "margin" here. A little wordy, but may "Past ice thickness for areas that have become ice free" be an alternative?

> We decided to stick with "margin" as this phrase is used throughout the manuscript and is defined in Section 3.3.

L. 85f: a) Please don't mix the notation $^{-1}$ and $/$. b) I believe this $/$ should not be here at all? (See Pattyn's Equation 11); From the equation above, I understand that this was set to n=3?

> We have corrected the notation as suggested.

L.86: I'm somewhat guessing but I imagine that, rather being "quantified", $B$ is "tuned" as to ensure that the flu solution matches the ice thickness.

> We changed the wording to "tuned" as suggested.

L. 92: I'm not sure to understand what this means.

> The "step in the elevation profile over ice cliffs" refers to the "steep elevation increase at the vertical ice cliffs".
> Reworded accordingly.

L.98f: I'm not sure: What was done in Experiment 1 then? I understood that this averaging happened in that experiment already? If not, what viscosity value were used for the locations at which there were no ice thickness observations?

> In Experiment 1 the generated margin thicknesses are used as thickness observations. During this experiment, the mean viscosity is generated within the reconstruction approach. This mean viscosity is in turn used in Experiment 2 as thickness input.
> Rephrased for clarity.

L. 100: What is the meaning of "generic data" here?

> Generic data refers to the margin thickness data.
> Reworded for clarity.

L. 109: As far as I'm concerned, this sentence an be removed.

> We decided to remove the sentence.

L. 124: Please clarify: is this wording referring to the results of Farinotti et al.?

> Yes, this phrase refers to results from Farinotti et al.
> Added source for clarity.

L. 127ff: I'm not sure to follow, is this discussion still referring to the "consensus thickness map"? Somehow, the focus seems to have shifted without noticing; Now I'm lost: What is this "second run" referring to? Is this Experiment 2? That's what the caption of Fig. 2 suggests. The wording is confusing.

> Added Experiment numbers for clarity.

L. 132ff: Please split this sentence in at least two parts. I apologize, but I could not follow.

> Split the sentence for easier readability.

L. 145: What is the meaning of "assimilated" here? Was the viscosity tuned again, as it was done for the ice thickness at the margin?

> For NIF, the GPR measurements are used as thickness input. This was referred to here.
> Rephrased for clarity.

L. 150: The concept of an "error margin" was not introduced, was it? I'm not sure to understand what is meant by that.

> Reworded for clarity.

L. 151: I'm again in the need of guessing: are these "separate entities" something defined by the RGI? Fig. 1 doesn't show three entities on NIF, though?

> The three different glacier entities are defined by the RGI and are also used in the Farinotti et al. consensus estimate. We have merged the three entities into one for our reconstruction, as the approach by Fürst et al. assigns the glacier margin an ice thickness of 0, which did not appear reasonable for the boundary lines between the different entities on the Northern Icefield.

L. 151ff: Sorry, I'm lost here: what is "model 1"? Is this meant to refer to Experiment 1 perhaps? This would be my first guess, but the next sentence is somewhat at odds with that. Has it something to do with "model 01" mentioned at L. 155?

> Model 1 (and later Model 01) refers to one of the models from the consensus estimate.
> Removed the single model data for more clarity and focused only on the consensus.

L. 159: Sorry, I'm lost again: didn't L. 155 say that the consensus has two models? Where is the third one coming from now, or why wasn't it mentioned at L. 155?

> The consensus estimate is made up of two model for NIF and three models for KG.
> Rephrased for clarity.

L. 166ff: Is my understanding correct: For NIF the consensus thickness is thus relatively close to both the GPR measurements and the results presented in this paper? This should probably be said explicitly as well, I imagine?

Yes, the mean ice thicknesses for the consensus estimate, our Experiments 2 and 3, as well as the reconstructions by Bohleber et al. are relatively close to one another. We have added a sentence stating this into the conclusion.

L. 176: Well, why is the volume never mentioned in the text then?

We replaced "volume" with thickness.

L. 181: What is the meaning of "retreat information"?

This refers to the lateral glacier retreat information, which was digitized from Landsat scenes and used in the margin thickness generation (Section 3.5)

L. 184f: Wait, didn't this "mean viscosity" come from the "margin ice thickness generated from DEMs and glacier outline" as well? How can this claim be made then?

The mean viscosity is generated from the margin ice thicknesses, but while local uncertainties from the margin thickness generation can influence the ice thickness distribution over the whole glacier (Fig. 2A KG), the mean viscosity shows a smoother ice thickness distribution, which seems more likely for Kersten Glacier (Fig. 2B KG). But as there are no thickness observations available for KG we cannot verify if the smoothed ice thickness distribution (Fig. 2B) is closer to reality or not.

L. 189f: I might be completely off track, but where would this mean viscosity come from at this stage? And isn't this claim somewhat in contradiction with what said at L. 135-136, i.e. that "the use of margin thickness information, generated from outline differences enabled a local glacier-specific viscosity tuning which might be preferential to an empirical temperature relation" (since, I assume, the latter would result in a mean viscosity as mentioned in the sentence)?

The mean viscosity is generated from within the thickness reconstruction approach. It is generated during the reconstruction using the margin thickness information generated from glacier outline differences. This means that by using glacier outline differences we can generate margin thickness information and then in a second step the mean viscosity. The results from our experiments show that for KG, where no ground/radar thickness observations were available, using the mean viscosity creates a smoother ice thickness distribution. This result might then be used preferential to approaches using empirical temperature relations to assess a glacier ice thickness as it is locally tuned from the direct glacier retreat as seen in satellite data.

**Reply to Reviewer RC2**

General Comments

In this study, the authors estimate the ice thickness of Northern Icefield and Kersten Glacier on Mt. Kilimanjaro in 2000 and 2011 using the ice thickness approach by Fürst et al (2017). Three different "experiments" are conducted to estimate the ice thickness, which either improves or are within the estimates from previous studies. The study makes good use of the few available observations, and the method and results are generally sound and interesting. I know this is a brief communication, but there are some key pieces of information that are missing within the data and model descriptions which I think are necessary to understand the manuscript, and the results would benefit from a discussion of uncertainties. In addition, the conclusion needs to be rewritten, as it does not seem to fit with the rest of

the paper. In general, the manuscript would also benefit from an increase in specificity and clarity, as I often had trouble following the text. I hope the technical comments are useful for improving this.

Specific Comments

L40: Location of the AWS is not on Figure 1. Also, for how long a period has the AWS measured and what components are measures (and with what uncertainties)?

> The AWS collected data from February 2005 to September 2013. It is located on Kersten Glacier at 5873 m.a.s.l. and the measurements include incoming and outgoing radiative fluxes (longwave and shortwave) with an accuracy of ±10%, air temperature (±0.2°C), relative humidity (±2% units), wind speed and wind direction (±0.3 ms$^{-1}$ and ±5°), air pressure (±0.2hPa) and the distance to the surface (±0.4%) (Section 3 c. in Mölg et al. 2009a).

> Due to the limited space in a "brief communication" we added a reference to the corresponding article by Mölg et al.

L48-49: This is not quite correct. The thickness estimate was created from the GPR by doing kringing interpolation, and it is an estimate from the whole area, not just the flat central part. Later in the manuscript (L 169), you seem to use the estimate that Bohleber created from the DEM, so I would mention that result here too. E.g. "In addition, ground penetrating radar (GPR) profiles from September 2015 (Fig. 1) were created by Bohleber et al. (2017). Using a kringing interpolation and the KILISoSDEM, the authors estimated the mean thickness to be between 21.2 ± 1 m and 27 ± 2 m."

> Implemented your suggestion into the manuscript.

L55-56: The Bohleber estimate is from 2015 and the consensus estimate is from 2000, so that should also contribute to the differences.

> Added a sentence concerning the different years for clarity.

L39-56: For most of the observations you do not provide uncertainty estimates

> We deliberately decided to not account for the input uncertainties in this case study as the focus is to exploit multi-temporal satellite information to better constrain a thickness reconstruction. Uncertainty consideration are covered in the methodological study by Fürst et al. (2017) comprising a spectrum of leave-out and sensitivity experiments. In light of the short communication format, it appears distracting to expand on the propagation of the input uncertainties into the final result. Moreover, input fields (SMB, DEM, outlines, thickness measurements) are often not necessarily provided with a robust error estimate. We therefore decided to refer the interested reader to Fürst et al. (2017) concerning the associated uncertainties.

L61: You have a point measurement in one location. How to you get distributed mass balance maps from one point on one glacier? And do you use the mean SMB from 2005-2013 for the 2000 and 2011 estimation? If you do, you should mention this as a possible source of uncertainty (and if you don't, how do you find the 2000 SMB?).

> The surface mass balance model (Mölg et al. 2008, 2009a) creates the distributed mass balance based on a DEM (we used the SRTM), which creates the lower boundary conditions for the

model and the meteorological data from the AWS, which are used as the model driver (Mölg et al. 2009a). We use the mean SMB for both, the 2000 and 2011 glacier states.

L61-65: Again, what do you use as forcing for NIF if the AWS is on KG? And how do you use the sonic ranger to test refreezing on NIF if it is mounted on KG? In addition, you should mention the sonic ranger in the data section and not only in the methods.

We use the meteorological data gathered by the AWS on KG as forcing on NIF as well. There is an WS installed on NIF from which we use a plotted time series of the sonic ranger measurements to which we compare our modelled accumulated surface height change. As the ice thickness reconstruction use the mean annual surface mass balance as input, we mainly used the total accumulated surface height change over the time period/ at the end of the modelling period (2013 September) to compare our results to. So the climatic variables (T, RH, ..) are the same for NIF and KG, but the topographic/elevation data differs, as this is directly calculated from the digital elevation model SRTM.

L87: What method do you use for interpolating?

The method used for interpolation is Natural Neighbor/Sibsonian Interpolation.

We mentioned the method now in the manuscript.

L109-122: A table with the different main thicknesses estimates would be useful and make comparison easier for the reader. I would e.g. include the mean thickness for each experiment, the mean thickness in the consensus estimate and Bohleber et al, and perhaps the thickness at the borehole locations. I know this is a brief communication and you are not allowed more figures, but maybe as a supplement.

We added a table containing the mean thickness estimates for NIF and KG and the thickness at the Thompson et al. (2002) borehole locations to the supplement (Supplementary Table 1).

L122: is it possible to calculate an uncertainty on the mean numbers? e.g. by leaving some GPR points out of the simulation and using those points for validation? Or if that would be too much work, you could give an approximation from the core location values (but then only for 2000). You already give it in percentage in the discussion, but here you could use the maximum absolute value.

We added the suggested approximation of absolute values at the core locations. For Experiment 1(2) the ice thickness at the core locations differ by 19.9 m (4.4 m) at C1, 23.9 m (8.3 m) at C2 and 36.6 m (26.1 m) at C3.

L124-174: The discussion would benefit from a short discussion on model uncertainties. For example for the constant viscosity runs, did you conduct a sensitivity analysis? Can you give an approximate uncertainty estimate of the SMB? And are there any uncertainties associated with the use of SIA?

The inferred **viscosity values** not only depend on the structural and temperature properties of the glacier body. They are also affected by the uncertainties of all other input fields and measurements. As the input uncertainty is already analyzed in depth by withholding GPR measurements in Fürst et al. (2017), it seemed redundant to repeat this exercise here. Certainly, in light of the short article format.

We cannot give an approximation on the **SMB uncertainty**, but as previously shown in Fürst et al. (2017) its influence on the ice thickness reconstruction is only minor. It was shown that, by changing the SMB input drastically, the mean ice thickness is reduced by 5% and the estimated ice volume by 4%. These values were found when the least amount of direct thickness measurements was assimilated. Moreover, this influence was estimated for various glacier

geometries, including an ice cap, on Svalbard. They also noted that, where ice thickness data is available, the influence of SMB input is compensated by direct observations (Fürst et al. 2017).

First of all, the **SIA** is a key component of this type of reconstruction. An expansion to include the solution of more complete forms of the force balance would require fundamental adjustments in the method. Though more complete, the problem might become even less well-posed and the computing requirements would increase unproportionate. Some of the uncertainties associated to the choice of the SIA are covered in Fürst et al. (2017).

L181-183: Why did you use a method which uses the SIA if the glacier is dynamically inactive? Would a plastic approach not be a better choice? Also, I think this section would fit better in the discussion.

We use the SIA in our reconstruction, as it is the method implemented in our reconstruction approach. KG is located on the steep flank of Mt. Kilimanjaro and we expect some ice motion. For NIF, this issue might be more relevant, and we expanded the discussion of this aspect in the revised manuscript. In such setups, the mass-conserving SIA approach is not ideal. We have no model to use a plastic flow assumption, so this approach was also not viable for us.

Concerning a plastic approach, it would certainly be an alternative here. Yet such approaches have often been applied in flowline setups with appropriate spatial averaging of the geometric input. Although one could theoretically apply them in 2D to each grid point, it would require an extra article to assess what the best strategies would be for spatial smoothing of the required input. We are unaware of a precursor study that applies the perfect plasticity concept in 2D (without final spatial interpolation) that is readily transferable to the complex topography of NIF.

L184-190: I was a bit puzzled on how you reach this conclusion. You suddenly mention "mean viscosity" experiments for NIF, although you did not mention this anywhere in the paper (Only for KG, as written in Table 1). For all three experiments, you always generated a viscosity field from observations for NIF (first from the margins, then using Bohleber et al data). You write that "the reconstructions reveal that if there are no thickness observations available, better results can be achieved with a mean viscosity value as input for ice thickness, instead of margin ice thickness generated from DEMs and glacier outline difference" but from what do you reach this conclusion? For KG you wrote the results for the margin method and the viscosity method were almost equal (and you use the margin method to get the mean viscosity in the first place), and for NIF you did not test it. Please clarify. And if you did do the mean viscosity test for NIF too, you should provide it in the paper.

We fear that we have not been careful enough in presenting the experiments which raised this concern. In the case of 'directly using lateral thickness information' and in the case of the 'mean viscosity', the thickness information from the retreat (ice-free area) is used. The difference is only how the reconstruction deals with this data. The two options are that the viscosity of each 'lateral thickness point' is used individually for an interpolation over the domain, resulting in a spatially variable ice viscosity (Experiment 1, NIF and KG). Otherwise, the viscosity point information is simply averaged, and a uniform value is used for the entire glacier (Experiment 2 and 3, KG).

L196-198: Wouldn't how well the margin method / mean viscosity method works depend on the size of the glacier?

We believe that the size of the glacier would most likely influence the outcome of the margin/mean viscosity method, but we have not tested the approach on glaciers of different sizes so we cannot comment on that further. The Kilimanjaro setup is quite special, and it is difficult to assess the glacier size dependence. Yet, glacier retreat is mostly expressed at low elevations. It is there that we expect to acquire past thickness values from multi-temporal satellite

information. As the frontal area represents an increasingly smaller portion of the entire system as glaciers become larger, the size dependence is certainly an interesting question. We can unfortunately not answer this here on the basis of the two very different glacier types on Mt. Kilimanjaro.

Technical Comments

L10: Add the thickness in 2000 too

We refrained from adding the 2000 thicknesses into the abstract as the word limit did not allow us to explain the difference (thickness increase) between the 2000 and 2011 reconstructions sufficiently and we believe it might cause confusion without the proper explanation.

L11: Write the unrealistically thick value

Changed the manuscript accordingly.

L11: change "meanwhile" to "have become"

Changed the manuscript accordingly.

L13: change "indicator" to "indicators"

Changed the manuscript accordingly.

L14: delete "As"

We decided to stick with this wording.

L20: delete "to"

Changed the manuscript accordingly.

L24: "assessment on" to "assessment of"

Changed the manuscript accordingly.

L25-28: You haven't introduced what you will do in this study yet, so a bit odd to talk about comparison already. I would suggest changing to: "A recent study attempted to reconstruct the distributed ice thickness for all glaciers outside of Antarctica using a consensus of up to 5 models (Farinotti et al. 2019). This estimate generated ice thicknesses estimates for Northern Icefield (NIF) and Kersten Glacier (KG) using ensembles of 2 and 3 models, respectively." Then at the end of line 37 you can add "The resulting thickness estimates are then compared with the consensus estimate" or similar.

Reworded the passage according to the suggestion.

L28-31: I would suggest dividing the sentence in two to make it easier to read: ". . . (Farinotti et al. 2019). In addition, it was recently discovered that KG has separated into two fragments, which is not in agreement with the estimated high thickness values in the study." I would also add a citation for the separation.

> Divided sentence as suggested. Added reference to the Landsat scene used in the study.

L34: I would suggest adding a line describing the model here, e.g. something like L80-83. Currently you mention a SMB model in L 39 without introducing that you even use it first.

> Added information on the SMB model in the introduction. As this manuscript is a "brief communication" we refrained from adding further information on the reconstruction approach in the introduction. We added a cross-reference to the corresponding section 3.4.

L39: either delete "the distributed surface mass balance (SMB) model and" or introduce the model in the introduction.

> We briefly introduced the model in the introduction.

L41: define DEM the first time you use it

> Added definition of DEM.

L41-43: missing reference for SRTM and Landsat 5

> We added references to the data sets used.

L43: change "from a merge of" to "by merging"

> Reworded to "by differencing from a merge of two …"

L46: reference Fig 1 after describing the redefinition

> Added reference to Figure 1.

L46: Future separation? Earlier you wrote it already separated?

> We anticipate a future separation of the Northern Icefield. Kersten Glacier has already separated. Reworded for clarification.

L47: delete "apart from" and add "were" before drilled

> Deleted words as suggested.

L48: can you add the borehole locations to figure 1 instead? It would be nice to have all the observations in the same figure.

Added borehole locations to Fig. 1 and removed them from Fig. 2.

L48: Definite GPR first time you use it

Defined acronym.

L54: change "showed a mean" to "had a mean"

Changed wording as suggested.

L54: give the value for NIF, "similar value" is too vague

Removed passage from manuscript.

L61-65: You should explain the reason for the model changes first, as it will be easier for the reader to follow. E.g. "The full MB model has only previously been verified for KG. However, because of the low slope angles of NIF, meltwater cannot run off from the surface of its planar top before refreezing sets in (Mölg and Hardy 2004), which was not captured by the model. Therefore we upgraded the model so that refreezing of meltwater is allowed on a bare ice surface with a slope angle below 5 degrees. With these changes, the model is capable of reproducing the observed surface height changes observed by a Sonic Ranger mounted to the AWS."

Rephrased the section for clarity with the suggestions in mind.

L76: change "nowadays" to "currently" or "2011"

Changed "nowadays" to "currently".

L89: change "increase" to "increased"

Changed word as suggested.

L90-91: I suggest changing the structure so the reasoning is before the how, e.g.: "In order to smooth the surface slope during reconstruction we use use the coupling length parameter, which is defined a a multiple of the local ice thickness."

Changed wording as suggested.

L95: add "by" before "combining"

Added the word "by" as suggested.

L98: the values are inferred and then the values are interpolated for the whole area?

We rephrase this passage and hope that it became clearer now.

L117: change "a distribution" to "the distribution"

Reworded the sentence.

L144: reference is missing a year

Added missing year to the reference. The reference is Thompson et al. 2002.

L147: what is "the better model"?

Removed the distinction of the two models that make up the consensus estimate for NIF for easier understanding and reworded the passage.

L149: change the end of the sentence to ".. the consensus estimate underestimates the the thickness at these points."

We rephrased a large part of the discussion for clarity, so the sentence referred to here was completely changed.

L165: mention the 10 and 5 m experiments in methods

Mentioned the 10 and 5 m experiments in the methods section 3.4.

"With the higher DEM quality in 2011, the resolution was iteratively increased from 25, via 10 and 5, to 2 m."

L169: remove "where the very high . . . as well"

We rephrased a large part of the discussion for clarity, so the sentence referred to here was completely changed.

L178: remove "became ice free or"

[revised manuscript text omitted]

hat formatiert: Schriftart: Nicht Fett